# Prior-Guided Diffusion Planning for Offline Reinforcement Learning

Donghyeon Ki[1]    JunHyeok Oh[1]    Seong-Woong Shim[1]    Byung-Jun Lee[1,2]

[1]Korea University    [2]Gauss Labs Inc.
{peop1e1n, the2ndlaw, ssw030830, byungjunlee}@korea.ac.kr

## Abstract

Diffusion models have recently gained prominence in offline reinforcement learning due to their ability to effectively learn high-performing, generalizable policies from static datasets. Diffusion-based planners facilitate long-horizon decision-making by generating high-quality trajectories through iterative denoising, guided by return-maximizing objectives. However, existing guided sampling strategies such as Classifier Guidance, Classifier-Free Guidance, and Monte Carlo Sample Selection either produce suboptimal multi-modal actions, struggle with distributional drift, or incur prohibitive inference-time costs. To address these challenges, we propose ***Prior Guidance*** (PG), a novel guided sampling framework that replaces the standard Gaussian prior of a behavior-cloned diffusion model with a learnable distribution, optimized via a behavior-regularized objective. PG directly generates high-value trajectories without costly reward optimization of the diffusion model itself, and eliminates the need to sample multiple candidates at inference for sample selection. We present an efficient training strategy that applies behavior regularization in latent space, and empirically demonstrate that PG outperforms state-of-the-art diffusion policies and planners across diverse long-horizon offline RL benchmarks. Our code is available at `https://github.com/ku-dmlab/PG`.

## 1 Introduction

Offline reinforcement learning (RL) allows agents to learn effective policies from fixed, pre-collected datasets, eliminating additional environment interaction. This paradigm is especially beneficial in scenarios where exploration is prohibitively expensive, unsafe, or impractical [1]. A core challenge in offline RL arises from distributional shifts, which can lead learned policies to favor out-of-distribution (OOD) actions due to overestimated value functions. To mitigate this issue, many existing methods incorporate behavior regularization to keep the learned policy close to the behavior policy that generated the data [1, 2, 3, 4].

Recently, diffusion-based planners have been proposed as a promising direction for offline RL. Leveraging their strengths in modeling long-range temporal dependencies and conditioning on auxiliary information [5, 6, 7, 8, 9, 10], diffusion models are well-suited for sequence modeling in complex decision-making tasks. By conditioning on future return surrogates, diffusion models can generate trajectories biased toward high-value behaviors, thereby demonstrating superior performance over prior approaches in long-horizon navigation and manipulation benchmarks across diverse offline RL settings [11, 12, 13, 14, 15].

Diffusion planners have employed various guided sampling strategies to steer trajectory generation toward high-return outcomes. Classifier Guidance (CG) [6] leverages a learned critic to directly guide the denoising process toward high-value trajectories. Classifier-Free Guidance (CFG) [16], in contrast, achieves conditioning by interpolating between conditional and unconditional diffusion

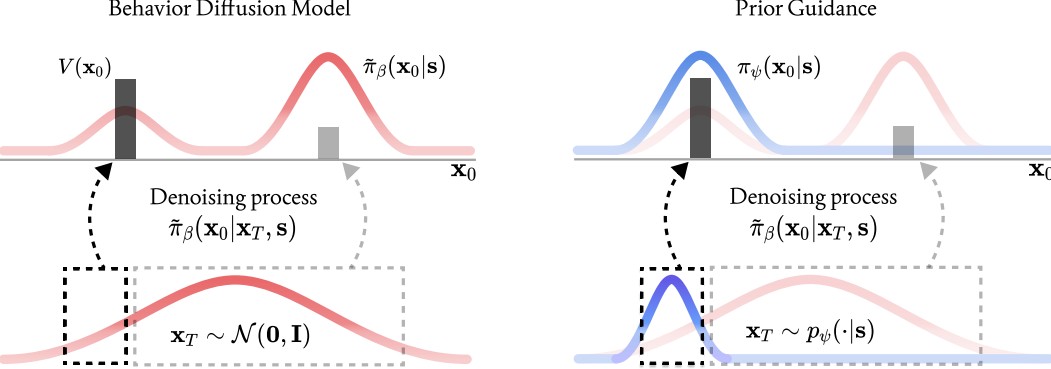

Figure 1: Comparison between the behavior-cloned diffusion model and Prior Guidance. **(Left)** The behavior diffusion model samples latent noise $\mathbf{x}_T$ from a standard Gaussian and generates trajectories $\mathbf{x}_0$ via a learned denoising process $\tilde{\pi}_\beta(\mathbf{x}_0|\mathbf{x}_T, \mathbf{s})$, approximating the dataset distribution. **(Right)** Prior Guidance instead samples $\mathbf{x}_T$ from a learned prior $p_\psi(\mathbf{x}_T|\mathbf{s})$ concentrated in high-value regions, leading to improved trajectory generation through the same denoising process.

models, removing the need for an additional network. While conceptually appealing, these methods face practical limitations: CG can produce suboptimal actions in settings where the optimal policy is multi-modal [17], and CFG often fails to generalize when the conditioning signal diverges from those seen during training, resulting in unreliable plans.

To address these challenges, recent high-performing diffusion planners have adopted a simple yet effective alternative—Monte Carlo sample selection (MCSS) [18, 19]. MCSS generates $N$ candidate trajectories and selects the one with the highest estimated return using a separately learned critic. While empirically successful, MCSS faces three primary limitations: (i) it incurs significant computational overhead at evaluation time due to the need to sample and score multiple trajectories, (ii) it does not guarantee the sampling of high-quality trajectories in the dataset, potentially overlooking valuable samples with high estimated value but low sampling likelihood, and (iii) increasing $N$ to improve coverage can lead to selecting out-of-distribution trajectories with spuriously high critic estimates.

Rather than guiding a pretrained diffusion model at inference time, an alternative approach is to directly train the diffusion model to generate high-value behaviors by maximizing a learned critic under behavior regularization [20, 21, 22, 23]. However, this training objective typically incurs substantial computational overhead, as it requires backpropagation through the entire denoising process. Consequently, these efforts have primarily focused on developing diffusion policies rather than diffusion planners. Moreover, since diffusion models do not offer tractable density evaluation, most existing methods have relied on simple forms of behavior regularization, such as incorporating auxiliary behavior cloning losses.

To this end, we propose ***Prior Guidance*** (PG), a novel guided sampling method that trains the prior distribution that decodes into high-value trajectories. After the training of a diffusion model through behavior cloning, we replace the standard Gaussian prior with a learnable parametrized prior distribution. This learnable prior is optimized using a behavior-regularized objective, encouraging the sampling of high-value trajectories through the diffusion model while remaining close to the behavior distribution (See Figures 1 and 2).

PG effectively addresses the limitations of previous methods. In contrast to MCSS, PG generates a single trajectory during inference, significantly reducing computational overhead. Compared to end-to-end training of the diffusion model with a behavior-regularized objective, PG is more efficient, as it avoids backpropagation through the full denoising process and requires optimizing only a small set of parameters. Furthermore, by modeling the target prior as a simple Gaussian distribution, PG enables more expressive and analytically tractable behavior regularization, achieving a more favorable trade-off between maximizing potential performance and mitigating the risk of out-of-distribution action selection. Empirically, PG outperforms existing diffusion-based methods and achieves state-of-the-art performance on long-horizon tasks in the D4RL offline RL benchmark suite [24].

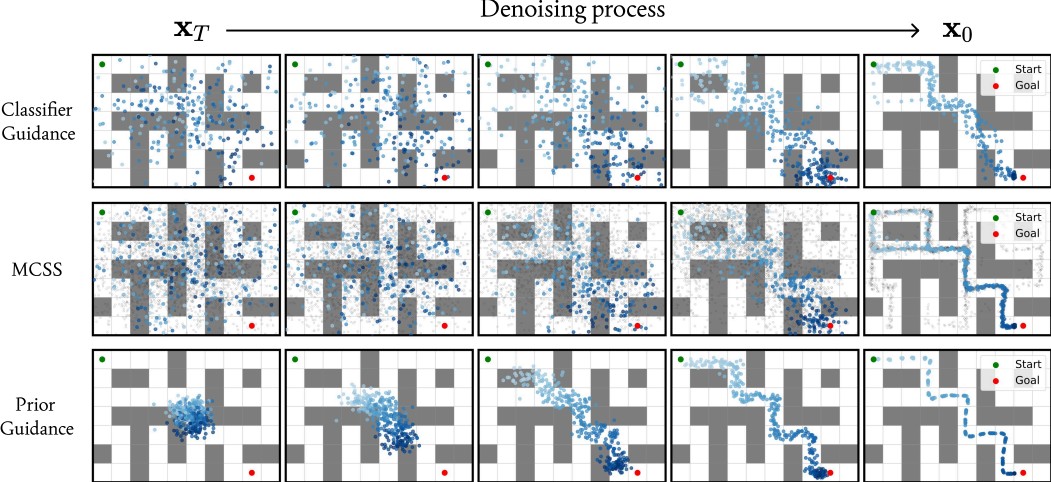

Figure 2: Visualization of trajectory generation using three different guided sampling methods on `maze2d-large-v1`. Each method produces 20 planned trajectories. For MCSS, each trajectory is selected from a set of 10 candidates ($N = 10$); the unselected candidates are visualized in gray.

## 2  Preliminaries

**Diffusion Probabilistic Models**   Diffusion models [5] are a class of generative models based on a Markovian forward–reverse process. The forward diffusion process gradually corrupts a data sample $\mathbf{x}_0$ by adding Gaussian noise over $T$ steps, defined as $q(\mathbf{x}_{1:T}|\mathbf{x}_0) = \prod_{t=1}^{T} q(\mathbf{x}_t|\mathbf{x}_{t-1})$, where $q(\mathbf{x}_t|\mathbf{x}_{t-1}) = \mathcal{N}(\mathbf{x}_t; \sqrt{1-\beta_t}\mathbf{x}_{t-1}, \beta_t\mathbf{I})$ and $\beta_t \in (0,1)$ is a predefined variance schedule. The reverse process is a parameterized Markov chain starting from a standard Gaussian prior $p(\mathbf{x}_T) = \mathcal{N}(\mathbf{0}, \mathbf{I})$, defined as $p_\theta(\mathbf{x}_{0:T}) = p(\mathbf{x}_T)\prod_{t=1}^{T} p_\theta(\mathbf{x}_{t-1}|\mathbf{x}_t)$, where $p_\theta(\mathbf{x}_{t-1}|\mathbf{x}_t) = \mathcal{N}(\mathbf{x}_{t-1}; \mu_\theta(\mathbf{x}_t, t), \Sigma_\theta(\mathbf{x}_t, t))$ and both $\mu_\theta$ and $\Sigma_\theta$ are predicted by a neural network.

The model is trained by minimizing a variational upper bound on the negative log-likelihood. In practice, a simplified objective is widely used, where the model learns to predict the added noise:

$$\mathcal{L}(\theta) = \mathbb{E}_{\mathbf{x}_0,\epsilon,t} \left[ \left\| \epsilon - \epsilon_\theta \left( \sqrt{\bar{\alpha}_t}\mathbf{x}_0 + \sqrt{1-\bar{\alpha}_t}\epsilon, t \right) \right\|^2 \right], \tag{1}$$

where $\epsilon \sim \mathcal{N}(\mathbf{0}, \mathbf{I})$ and $\bar{\alpha}_t = \prod_{i=1}^{t}(1-\beta_i)$. This training objective corresponds to denoising score matching across multiple noise levels. Once trained, the model can generate new samples by iteratively following the reverse process defined above. Alternatively, [25] proposed another generative process that shares the same marginal distribution, defined by

$$\mathbf{x}_{t-1} = \sqrt{\bar{\alpha}_{t-1}} \left( \frac{\mathbf{x}_t - \sqrt{1-\bar{\alpha}_t}\epsilon_\theta(\mathbf{x}_t, t)}{\sqrt{\bar{\alpha}_t}} \right) + \sqrt{1-\bar{\alpha}_{t-1}-\sigma_t^2} \cdot \epsilon_\theta(\mathbf{x}_t, t) + \sigma_t\boldsymbol{\epsilon},$$

where $\epsilon \sim \mathcal{N}(\mathbf{0}, \mathbf{I})$ and $\sigma_t$ is a tunable parameter that controls the stochasticity of the generation. When $\sigma_t = 0$, the generative process becomes entirely deterministic, resulting in the Denoising Diffusion Implicit Model (DDIM) sampling scheme.

**Offline Reinforcement Learning**   We assume the reinforcement learning problem under a Markov Decision Process (MDP). The MDP can be represented as a tuple $\mathcal{M} = \{S, A, P, R, \gamma, p_0\}$, where $S$ is the state space, $A$ is the action space, $P(\mathbf{s}'|\mathbf{s}, \mathbf{a}) : S \times A \to \Delta(S)$ is a transition dynamics, $R(\mathbf{s}, \mathbf{a}) : S \times A \to \mathbb{R}$ is a reward function, $p_0 \in \Delta(S)$ is an initial state distribution and $\gamma \in [0, 1)$ is a discount factor. Offline RL aims to train a parameterized policy $\pi_\psi(\mathbf{a}|\mathbf{s})$, which maximizes the expected discounted return from a dataset $\mathcal{D}$ without environment interactions.

A common strategy in offline RL is to optimize policies via a behavior-regularized objective: $\max_\psi \mathbb{E}_{\mathbf{s}\sim\mathcal{D},\mathbf{a}\sim\pi_\psi(\cdot|\mathbf{s})} \left[ Q(\mathbf{s}, \mathbf{a}) - \alpha f\left( \frac{\pi_\psi(\mathbf{a}|\mathbf{s})}{\pi_\beta(\mathbf{a}|\mathbf{s})} \right) \right]$, where $Q$ denotes the action-value function of policy $\pi_\psi$, $\pi_\beta$ is the behavior policy, and $f(\cdot)$ is a regularization function (e.g., for $f$-divergence) [2, 4, 26, 27, 28]. However, when employing a diffusion policy or planner to model

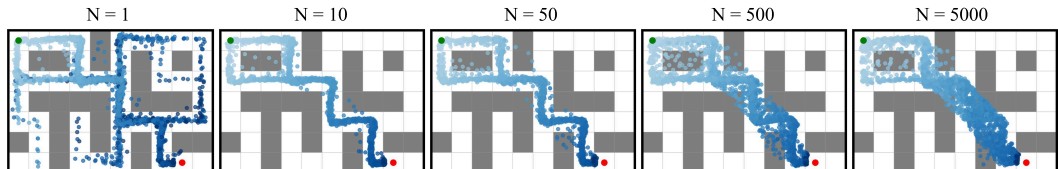

Figure 3: Predicted state sequences in `maze2d-large-v1` based on MCSS varying the number of candidate trajectories $N$. When $N$ is small ($N = 1$), MCSS often fails to sample trajectories with high value estimates, as the likelihood of capturing an optimal trajectory within a limited set is low. Conversely, when $N$ is excessively large ($N = 500$ or $5000$), the method tends to select OOD trajectories with inflated value estimates.

$\pi_\psi$, directly evaluating the density $\pi_\psi(\mathbf{a}|\mathbf{s})$ becomes intractable, making it challenging to apply sophisticated regularization functions. As a result, prior works have instead resorted to using a diffusion behavior cloning loss (1) as a surrogate for behavior regularization [20, 21, 22, 23].

**Diffusion Planner**   Diffusion-based planners [11, 12, 13, 14, 15, 19] generate a trajectory of length $H$ starting from the current time step $\tau$. While various design choices are available, in this work, we largely follow the recommendations of [19], which are based on comprehensive experimental analysis. Specifically, a diffusion planner is trained to generate $H$-length sequences of current and future states $\mathbf{x}^{(\tau)} = [\mathbf{s}^{(\tau)}, \mathbf{s}^{(\tau+m)}, \ldots, \mathbf{s}^{(\tau+(H-1)m)}]$ with the planning stride of $m$, where $\tau$ denotes the environment timestep. To bias the generation toward high-return behaviors, a trajectory critic $V(\mathbf{x}^{(\tau)}) = \mathbb{E}\left[\sum_{h=0}^{\infty} \gamma^h r^{(\tau+h)} | \mathbf{x}^{(\tau)}\right]$ is estimated and utilized during planning. This critic can be either the behavior value function or the target value function. After producing a high-value state sequence $\mathbf{x}^{(\tau)}$, a separately trained inverse dynamics model is employed to recover the corresponding action $\mathbf{a}^{(\tau)}$ from the predicted sequence. For simplicity, we overload the notation and denote the diffusion planner-based policy as $\pi(\mathbf{x}|\mathbf{s})$.

A **detailed discussion of related work** is provided in the Appendix A.

## 3   Diffusion Planning with Prior Guidance

**Limitations of Monte Carlo Sample Selection (MCSS)**   Monte Carlo Sample Selection (MCSS), which selects the best sample from $N$ candidates based on a separately trained value function, has been shown to achieve superior performance and has been widely adopted in recent diffusion-based offline RL [17, 18, 19]. Nonetheless, as discussed above, MCSS also exhibits significant limitations: (i) it incurs substantial computational overhead during inference, as it requires sampling multiple candidates from diffusion models, which are already expensive to sample from, (ii) it does not guarantee coverage of high-value samples within the behavioral distribution, and (iii) while a sufficiently large $N$ is necessary for effective guidance, excessively large $N$ increases the risk of selecting OOD samples with overestimated value estimates, leading to performance degradation.[1] Figure 3 illustrates these limitations.

### 3.1   Prior Guidance

To mitigate these shortcomings, we aim to adopt a behavior regularization framework in providing guidance for diffusion planners. Concretely, the objective is formalized as:

$$\max_\psi \mathbb{E}_{\mathbf{s}\sim\mathcal{D}, \mathbf{x}_0\sim\pi_\psi(\cdot|\mathbf{s})} \left[ V(\mathbf{x}_0) - \alpha f\left( \frac{\pi_\psi(\mathbf{x}_0|\mathbf{s})}{\pi_\beta(\mathbf{x}_0|\mathbf{s})} \right) \right], \tag{2}$$

where $\mathbf{x}_0$ denotes a fully denoised trajectory generated by the diffusion model. This objective promotes the generation of high-value trajectories while constraining them to stay close to the dataset.

---

[1]A notable variant is SfBC [29], which performs resampling by assigning weights proportional to $\exp(\alpha Q(\mathbf{s}, \mathbf{a}))$ over a set of $N$ candidate actions generated by a diffusion policy. This weighting scheme implicitly induces KL-behavior regularization and mitigates limitation (iii). However, it introduces an additional hyperparameter $\alpha$, increasing the complexity of hyperparameter tuning. Moreover, due to other unresolved limitations, the method continues to incur significant inference-time computational overhead.

However, directly optimizing this objective with diffusion models poses significant challenges: (i) it requires expensive backpropagation through the iterative denoising process, which becomes particularly costly for high-dimensional diffusion planners; and (ii) evaluating $\pi_\psi(\mathbf{x}_0|\mathbf{s})$ is non-trivial, as diffusion models do not admit tractable density estimation.

We now introduce ***Prior Guidance***, a method for guiding diffusion models with a learnable prior distribution, which is free from the abovementioned challenges. Let $\tilde{\pi}_\beta$ denote a diffusion planner trained via behavior cloning. Given a state $\mathbf{s}$, the model generates a trajectory $\mathbf{x}_0$ by first sampling an initial noise $\mathbf{x}_T \sim p(\mathbf{x}_T) = \mathcal{N}(\mathbf{0}, \mathbf{I})$, followed by a denoising process. Assuming a DDIM sampling, the conditional density $\tilde{\pi}_\beta(\mathbf{x}_0|\mathbf{s})$ can be expressed as:

$$\tilde{\pi}_\beta(\mathbf{x}_0|\mathbf{s}) = \mathbb{E}_{p(\mathbf{x}_T)}\left[\delta(\mathbf{x}_0 = g_\mathbf{s}(\mathbf{x}_T))\right],$$

where $\delta(\cdot)$ denotes the Dirac delta function, and $g_\mathbf{s}$ is the deterministic denoising operator of the behavior diffusion model $\tilde{\pi}_\beta$ for state $\mathbf{s}$.

[25] has demonstrated that interpolations in the latent space $\mathbf{x}_T$ yield smooth and semantically meaningful trajectories in the sample space, indicating that $\mathbf{x}_T$ captures the semantic structure of $\mathbf{x}_0$. This observation motivates shifting where the learning happens from the diffusion model to the prior distribution. Accordingly, we propose to learn a parameterized Gaussian prior $p_\psi(\mathbf{x}_T|\mathbf{s})$, while keeping the denoising operator $g_\mathbf{s}$ fixed. The trajectory distribution after planning $\pi_\psi$ can then be reformulated as:

$$\pi_\psi(\mathbf{x}_0|\mathbf{s}) = \mathbb{E}_{p_\psi(\mathbf{x}_T|\mathbf{s})}\left[\delta(\mathbf{x}_0 = g_\mathbf{s}(\mathbf{x}_T))\right].$$

When a sufficiently large number of discretization steps are employed, the denoising operator $g_\mathbf{s}$ can be viewed as an approximate, bijective ODE solver. This bijection allows us to re-express previously intractable behavior-regularized objective of the diffusion model in terms of a tractable objective over prior distributions:

**Proposition 1.** *Assume that DDIM sampling employs a sufficiently large number of discretization steps ensuring that the mapping $\mathbf{x_0} = g_\mathbf{s}(\mathbf{x}_T)$ is bijective. If the target trajectory density $\pi_\psi$ is parametrized by placing the prior $p_\psi(\mathbf{x}_T|\mathbf{s})$ while keeping the denoising process $g_\mathbf{s}$ fixed, the behavior-regularized objective* (2) *is equivalent to:*

$$\max_\psi \mathbb{E}_{\mathbf{s}\sim\mathcal{D}, \mathbf{x}_T\sim p_\psi(\cdot|\mathbf{s})}\left[V\left(g_\mathbf{s}\left(\mathbf{x}_T\right)\right) - \alpha f\left(\frac{p_\psi(\mathbf{x}_T|\mathbf{s})}{p(\mathbf{x}_T)}\right)\right]. \tag{3}$$

The proof is in the Appendix B. Note that the behavior regularization term $f(\cdot)$ is now evaluated between two Gaussian distributions ($p_\psi$ and $p$), rather than between intractable diffusion model densities, which enables closed-form computation under several popular formulations of $f$.

**Avoiding Backpropagation Through the Denoising Process**    The reformulated objective (3) still depends on $g_\mathbf{s}$, requiring costly backpropagation through the denoising process. To avoid this, we introduce a value function $\bar{V}$ that operates directly in the latent prior space $\mathbf{x}_T$ rather than on the denoised output $\mathbf{x}_0 = g_\mathbf{s}(\mathbf{x}_T)$. Due to the bijectivity of $g_\mathbf{s}$ and the smoothness of the latent space induced by the diffusion model, the latent value function $\bar{V}$ is well-defined and straightforward to learn. It is trained by minimizing the following objective:

$$\min_\phi \mathbb{E}_{\mathbf{x}_T\sim p_\psi(\cdot|\mathbf{s})}\left[\left(\bar{V}_\phi(\mathbf{x}_T) - V\left(g_\mathbf{s}(\mathbf{x}_T)\right)\right)^2\right], \tag{4}$$

where $V(g_\mathbf{s}(\mathbf{x}_T))$ serves as the target signal. Once trained, the latent space value function $\bar{V}_\phi$ is used to replace the original critic in the behavior-regularized objective (3):

$$\max_\psi \mathbb{E}_{\mathbf{s}\sim\mathcal{D}, \mathbf{x}_T\sim p_\psi(\cdot|\mathbf{s})}\left[\bar{V}_\phi(\mathbf{x}_T) - \alpha f\left(\frac{p_\psi(\mathbf{x}_T|\mathbf{s})}{p(\mathbf{x}_T)}\right)\right]. \tag{5}$$

This training scheme completely eliminates the need for gradient flow through the denoising operator $g_\mathbf{s}$. Gradients are propagated only through the prior distribution $p_\psi$ and the value function $\bar{V}_\phi$, while $g_\mathbf{s}$ remains fixed. Consequently, PG circumvents the major computational bottleneck associated with directly optimizing diffusion models for high-value behavior generation, thereby enabling its application to diffusion planners, whereas previous approaches were constrained to diffusion policies due to their computational overhead. In practice, we alternate between minimizing the regression loss in Eq. (4) and optimizing the prior via Eq. (5). In Section 4.2, we qualitatively analyze the latent value function $\bar{V}$ and show that it accurately captures the intended value structure.

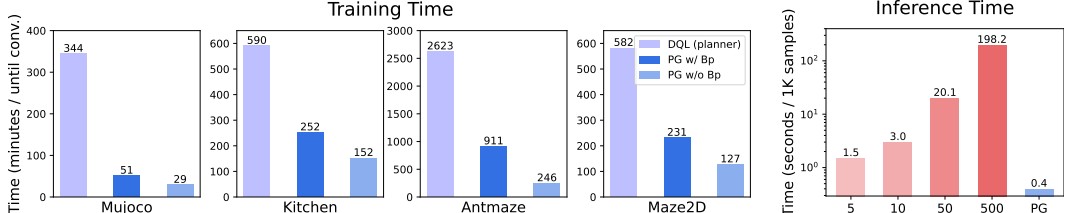

Figure 4: Training and inference time comparison. **(Left)** Training time until convergence for DQL (planner), PG with and without backpropagation through the denoising process across different environments. DQL (planner) denotes a baseline where the diffusion model is directly trained via Eq. (2); due to the intractability of computing the model's density, Eq. (1) is used as a surrogate. **(Right)** Inference time for MCSS and PG. The inference time is measured on the `antmaze-medium-play-v2`, with MCSS evaluated under different numbers of sampled trajectories.

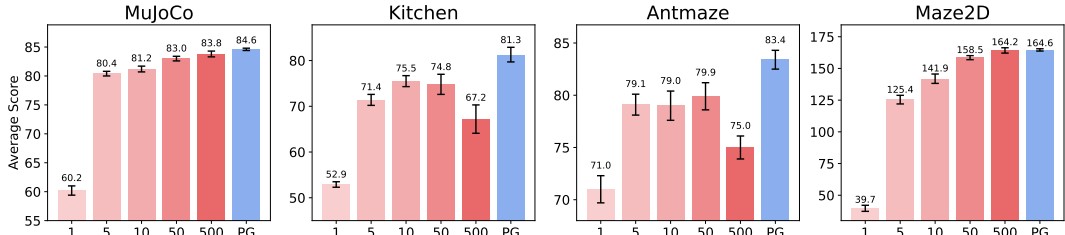

Figure 5: Average normalized score comparison between MCSS with varying numbers of samples $N$ and PG across four D4RL tasks. PG consistently matches or outperforms MCSS while requiring significantly fewer samples. Full results are provided in the Appendix D.

## 3.2 Does PG Address the Limitations of MCSS?

We empirically validate whether PG indeed resolves the limitations of MCSS, as claimed above. To ensure a fair comparison, all other experimental settings were held constant when comparing PG and MCSS.

**(1) Reduced computational cost during inference.** MCSS requires sampling multiple candidate trajectories at each environment step and selecting the one with the highest estimated value using a critic. Since sampling from a diffusion model is inherently computationally expensive, generating multiple candidates introduces substantial overhead, particularly in high-dimensional or long-horizon tasks. In contrast, PG generates a single trajectory starting from a learned prior, thereby avoiding this computational burden. Although PG introduces the additional cost of training the prior, it circumvents the need for costly backpropagation through the full denoising process by training a latent space value function, enabling efficient training.

As shown in Figure 4, PG w/o Bp—our proposed method that removes backpropagation through the denoising process—significantly reduces training time compared to PG w/ Bp. Moreover, PG achieves substantially lower inference time than MCSS, especially when the number of sampled trajectories $N$ is large, highlighting its superior computational efficiency. We also report the training time of Diffusion Q-learning when implemented as a planner-based method, denoted as DQL (planner). As expected, DQL (planner) is significantly more time-consuming to train than PG, taking up to ten times longer in the AntMaze domain. More experimental details for DQL (planner) are provided in the Appendix C.1.

**(2) Tractable and stable behavior regularization.** MCSS lacks an explicit mechanism to guide sampling toward trajectories aligned with the behavioral distribution, often resulting in OOD trajectory selections with spuriously high value estimates. PG addresses this limitation by performing regularization directly in the prior space, where the divergence between the learned prior $p_\psi$ and the standard Gaussian prior $p$ admits a closed-form expression. This enables stable and analytically tractable behavior regularization, which was previously infeasible in diffusion models due to the intractability of their output densities.

Table 1: Normalized scores on the D4RL benchmark. For each dataset, the best-performing method is highlighted in red. Prior Guidance (PG) consistently achieves strong performance, particularly on long-horizon tasks, and outperforms DV* in nearly all environments. We compute the mean and standard error over 5 random seeds. Due to space constraints, the standard error for all algorithms are provided in the Appendix E.

| Dataset | Gaussian policies | | | Diffusion policies | | | | | Diffusion planners | | | | | |
|---|---|---|---|---|---|---|---|---|---|---|---|---|---|---|
| | BC | CQL | IQL | SfBC | DQL | IDQL | QGPO | SRPO | Diffuser | AD | DD | HD | DV* | PG |
| Walker2d-M | 6.6 | 79.2 | 78.3 | 77.9 | 87.0 | 82.5 | 86.0 | 84.4 | 79.6 | 84.4 | 82.5 | 84.0 | 79.5 | 82.3 |
| Walker2d-M-R | 11.8 | 26.7 | 73.9 | 65.1 | 95.5 | 85.1 | 84.4 | 84.6 | 70.6 | 84.7 | 75.0 | 84.1 | 83.5 | 83.7 |
| Walker2d-M-E | 6.4 | 111.0 | 109.6 | 109.8 | 110.1 | 112.7 | 110.7 | 114.0 | 106.9 | 108.2 | 108.8 | 107.1 | 109.0 | 109.4 |
| Hopper-M | 29.0 | 58.0 | 66.3 | 57.1 | 90.5 | 65.4 | 98.0 | 95.5 | 74.3 | 96.6 | 79.3 | 99.3 | 84.1 | 97.5 |
| Hopper-M-R | 11.3 | 48.6 | 94.7 | 86.2 | 101.3 | 92.1 | 96.9 | 101.2 | 93.6 | 92.2 | 100.0 | 94.7 | 91.3 | 91.3 |
| Hopper-M-E | 111.9 | 98.7 | 91.5 | 108.6 | 111.1 | 108.6 | 108.0 | 100.1 | 103.3 | 111.6 | 111.8 | 115.3 | 109.9 | 110.4 |
| HalfCheetah-M | 36.1 | 44.4 | 47.4 | 45.9 | 51.1 | 51.0 | 54.1 | 60.4 | 42.8 | 44.2 | 46.7 | 50.9 | 45.6 | 45.6 |
| HalfCheetah-M-R | 38.4 | 46.2 | 44.2 | 37.1 | 47.8 | 45.9 | 47.6 | 51.4 | 37.7 | 38.3 | 39.3 | 38.1 | 46.4 | 46.4 |
| HalfCheetah-M-E | 35.8 | 62.4 | 86.7 | 92.6 | 96.8 | 95.9 | 93.5 | 92.2 | 88.9 | 89.6 | 90.6 | 92.5 | 92.3 | 95.2 |
| **Average** | 31.9 | 63.9 | 77.0 | 75.6 | 87.9 | 82.1 | 86.6 | 87.1 | 77.5 | 83.3 | 81.8 | 84.6 | 83.0 | 84.6 |
| Kitchen-M | 47.5 | 51.0 | 51.0 | 45.4 | 62.6 | 66.5 | – | – | 52.5 | 51.8 | 65.0 | 71.7 | 73.3 | 74.6 |
| Kitchen-P | 33.8 | 49.8 | 46.3 | 47.9 | 60.5 | 66.7 | – | – | 55.7 | 55.5 | 57.0 | 73.3 | 76.2 | 88.0 |
| **Average** | 40.7 | 50.4 | 48.7 | 46.7 | 61.6 | 66.6 | – | – | 54.1 | 53.7 | 61.0 | 72.5 | 74.8 | 81.3 |
| Antmaze-M-P | 0.0 | 14.9 | 71.2 | 81.3 | 76.6 | 84.5 | 83.6 | 80.7 | 6.7 | 12.0 | 8.0 | – | 80.8 | 87.8 |
| Antmaze-M-D | 0.0 | 15.8 | 70.0 | 82.0 | 78.6 | 84.8 | 83.8 | 75.0 | 2.0 | 6.0 | 4.0 | 88.7 | 82.0 | 87.3 |
| Antmaze-L-P | 0.0 | 53.7 | 39.6 | 59.3 | 46.4 | 63.5 | 66.6 | 53.6 | 17.3 | 5.3 | 0.0 | – | 80.8 | 82.4 |
| Antmaze-L-D | 0.0 | 61.2 | 47.5 | 45.5 | 56.6 | 67.9 | 64.8 | 53.6 | 27.3 | 8.7 | 0.0 | 83.6 | 76.0 | 76.0 |
| **Average** | 0.0 | 36.4 | 57.1 | 67.0 | 64.6 | 75.2 | 74.7 | 65.7 | 13.3 | 8.0 | 3.0 | – | 79.9 | 83.4 |
| Maze2D-U | 3.8 | 5.7 | 47.4 | 73.9 | – | 57.9 | – | – | 113.9 | 135.1 | – | 128.4 | 133.8 | 139.2 |
| Maze2D-M | 30.3 | 5.0 | 34.9 | 73.8 | – | 89.5 | – | – | 121.5 | 129.9 | – | 135.6 | 144.1 | 159.5 |
| Maze2D-L | 5 | 12.5 | 58.6 | 74.4 | – | 90.1 | – | – | 123 | 167.9 | – | 155.8 | 197.6 | 195.2 |
| **Average** | 13.0 | 7.7 | 47.0 | 74.0 | – | 79.2 | – | – | 119.5 | 144.3 | – | 139.9 | 158.5 | 164.6 |

Empirical results presented in Figure 5 corroborate this advantage: while MCSS exhibits a trade-off between trajectory optimality and OOD avoidance, PG with appropriate behavior regularization consistently outperforms MCSS with the best performing $N$ across diverse domains.

# 4 Experiments

**Practical Implementation** The target prior $p_\psi$ in PG is parameterized using a GRU [30] to capture the temporal dependencies in $\mathbf{x}$. Conditioned on the current state, the prior network outputs a sequence of mean and log standard deviation vectors that define independent Gaussian distributions over the latent noise dimensions. The current state is first provided as input to the GRU, and its output (prior to sampling) is fed back as input for subsequent steps. Detailed experimental settings are provided in the Appendix C.2.

Apart from the learnable prior, PG is built on top of Diffusion Veteran (DV) [19], following their design choices. Due to the long training time of the original implementation, we re-implemented DV and refer to this version as DV*. PG inherits from DV* the planner $g_\mathbf{s}$, the inverse dynamics

---

**Algorithm 1 Prior Guidance**

**Input:** Dataset $D$, Hyperparameter $\alpha$
**Require:** Planner $g_\mathbf{s}$, Inverse Dynamics $\epsilon_\omega$, Critic $V$
**Initialize:** Prior $p_\psi$, Critic $\bar{V}_\phi$

1: // Training
2: **for** each iteration **do**
3:     Sample $\mathbf{s} \sim D$ and $\mathbf{x}_T \sim p_\psi(\cdot \mid \mathbf{s})$
4:     Update critic $\bar{V}_\phi$ using Eq. (4)
5:     Update prior $p_\psi$ using Eq. (5)
6: **end for**
7: // Execution
8: $\mathbf{s}^{(0)} = $ env.reset()
9: **for** environment step $\tau = 0, 1, \dots$ **do**
10:     Sample $\mathbf{x}_T \sim p_\psi(\cdot \mid \mathbf{s}^{(\tau)})$
11:     Generate $\mathbf{x}_0$ using $g_\mathbf{s}$
12:     Predict action $\mathbf{a}^{(\tau)}$ using $\epsilon_\omega(\mathbf{x}_0)$
13:     $\mathbf{s}^{(\tau+1)} = $ env.step($\mathbf{a}^{(\tau)}$)
14: **end for**

---

model $\epsilon_\omega$, and the critic $V$ as described in Algorithm 1; following DV, PG uses the behavior value function for its critic $V$. The pseudocode for DV* is provided in the Appendix D, along with a detailed comparison to the performance of the original DV implementation.

## 4.1 D4RL Benchmark

We conducted experiments on the D4RL offline RL benchmark [24], which span a wide range of domains and dataset settings. We evaluate Prior Guidance (PG) across four representative tasks:

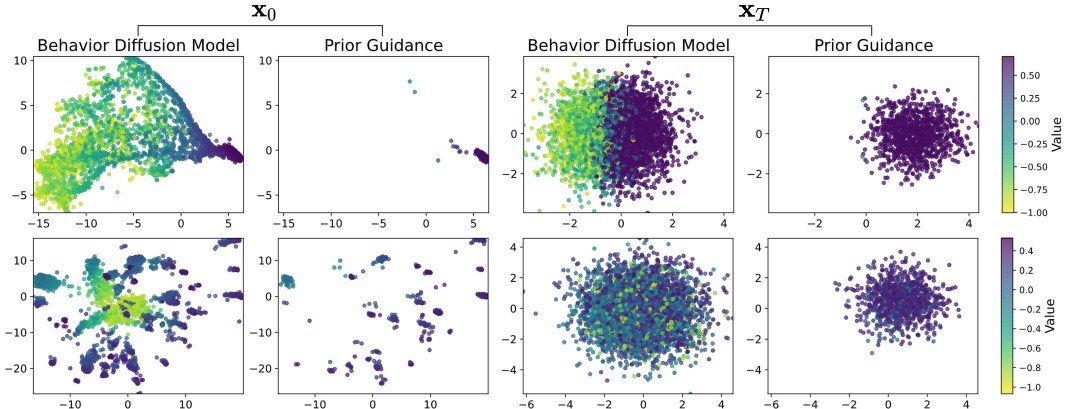

Figure 6: Visualizations of trajectories $(\mathbf{x}_0)$ and priors $(\mathbf{x}_T)$ from the behavior diffusion model and Prior Guidance in `maze2d-large-v1` (**top**) and `antmaze-medium-diverse-v2` (**bottom**). We applied PCA [34] to project the high-dimensional trajectories into two dimensions, with the color of each sample representing its estimated value. We plotted 5,000 samples from the standard normal prior of the behavior diffusion model and 1,000 samples from the target prior of PG.

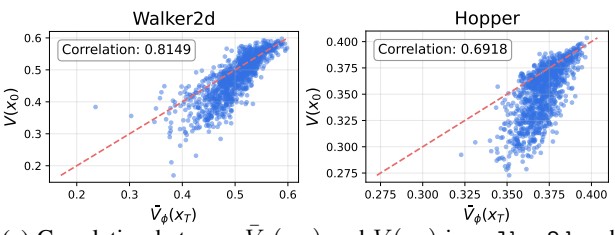

(a) Correlation between $\bar{V}_\phi(\mathbf{x}_T)$ and $V(\mathbf{x}_0)$ in `walker2d` and `hopper` environments.

| Dataset | PG w/ Bp | PG w/o Bp |
|---------|----------|-----------|
| MuJoCo | **85.8**±0.3 | 84.6±0.2 |
| Kitchen | 81.0±2.4 | **81.3**±1.6 |
| Antmaze | **84.0**±1.2 | 83.4±0.9 |
| Maze2D | 164.2±1.5 | **164.6**±0.9 |
| **Total** | **415.0** | 413.9 |

(b) Average scores with and without back-propagation through the denoising process.

Figure 7: Experiments to evaluate whether the latent value function $\bar{V}$ is accurately trained.

MuJoCo, Kitchen, Antmaze, and Maze2D. In MuJoCo tasks, the dataset types are denoted as M (medium), R (replay), and E (expert); in Kitchen tasks, M (mixed) and P (partial); in Antmaze tasks, M (medium), L (large), P (play), and D (diverse); in Maze2D tasks, U (umaze), M (medium), and L (large). We benchmark against BC, CQL [31], IQL [32], SfBC [29], DQL [20], IDQL [18], QGPO [33], SRPO [28], Diffuser [11], AD [13], DD [12], HD [15], and DV [19]. The compared methods are categorized into Gaussian policies (CQL, IQL), diffusion policies (SfBC, DQL, IDQL, QGPO, SRPO), and diffusion planners (Diffuser, AD, DD, HD, DV). The normalized scores are summarized in Table 1.

As shown in Table 1, PG achieves state-of-the-art performance on tasks that require long-horizon decision making. By addressing the key limitations of MCSS, it significantly improves upon DV*. In MuJoCo tasks where complex decision making is not required, diffusion planners generally underperform compared to diffusion policies. Although PG also exhibits relatively lower performance in these tasks, it achieves the highest scores among diffusion planners. In long-horizon tasks such as Kitchen, AntMaze, and Maze2D, PG substantially outperforms all baselines across most tasks.

## 4.2 Analysis

**Visualization on Latent Trajectories** We visualize the latent noise and corresponding trajectories from the behavior diffusion model and PG to verify whether the learned prior $p_\psi$ in PG accurately captures the noise $\mathbf{x}_T$ that is denoised into high-value trajectories $\mathbf{x}_0$. As shown in Figure 6, the prior of the behavior diffusion model spans a wide range of values, whereas the learned prior of PG predominantly samples noise that leads to high-value trajectories. This demonstrates that the latent space of the learned behavior diffusion model is structured sufficiently well to cluster high-value trajectories under a simple Gaussian prior. Although the target prior in PG remains close to

the standard normal distribution due to behavior regularization, it still captures mostly high-value trajectories.

**Consequence of Using Latent-Space Value Function**   To avoid backpropagation through the denoising process, PG trains a latent-space value function $\bar{V}$ directly on the prior noise $\mathbf{x}_T$ (4). Figure 7 presents our empirical evaluation of this design choice. In particular, Figure 7a depicts the correlation between learned latent values $\bar{V}_\phi(\mathbf{x}_T)$ and ground-truth values $V(\mathbf{x}_0)$ for the `walker2d-medium-v2` and `hopper-medium-v2` tasks. While a strong positive correlation is observed, a perfect learning of the mapping would yield a correlation of exactly 1. The observed deviation reflects approximation errors, which could potentially affect performance when relying on the latent value function. However, our experiments show that these approximation errors do not lead to actual performance degradation. As shown in Figure 7b, PG achieves nearly identical normalized scores on the D4RL benchmark with and without backpropagation through the denoising process, confirming that training $\bar{V}$ in latent space is sufficient. Full results corresponding to Figure 7b are provided in the Appendix G.

**More Expressive Target Prior**   To investigate the flexibility and representational capacity of PG, we extend our approach to use a multi-modal Gaussian mixture prior in place of the standard uni-modal Gaussian prior. This modification allows us to evaluate whether a more expressive prior distribution can yield performance gains. As shown in Table 2, the results

Table 2: Average normalized scores of uni-modal and multi-modal Gaussian prior on MuJoCo and Antmaze tasks.

| Dataset | uni-modal | multi-modal |
|---------|-----------|-------------|
| MuJoCo  | **84.6**±0.2 | 82.4±0.4 |
| Antmaze | 83.4±0.9 | **85.8**±0.6 |

are domain-dependent: in MuJoCo tasks, the uni-modal Gaussian prior slightly outperforms the multi-modal variant, whereas in AntMaze tasks, the multi-modal prior provides marginal improvements. These findings suggest that the benefit of a more expressive prior is more pronounced in complex, long-horizon environments such as AntMaze, while in simpler control tasks like MuJoCo, the latent space found by a diffusion model is sufficiently well structured to capture high-value trajectories only with a uni-modal gaussian distribution. Full results corresponding to Table 2 are provided in the Appendix G.

### 4.3   Ablation Studies

**Regularization Function** $f$   By deriving a closed-form solution for behavior regularization, PG accommodates a variety of regularization functions $f$. To assess the impact of different choices of $f$, we performed an ablation study on the D4RL benchmark. We compared $f(x) = -\frac{\log x}{x}$ (KL-divergence), $f(x) = \log x$ (reverse KL-divergence), and $f(x) = \frac{(x-1)^2}{x}$ (Pearson $\chi^2$-divergence). In practice, we use KL-divergence in PG. Ablation results are provided in the Appendix H.

**Behavior Regularization Coefficient** $\alpha$   We present ablation studies evaluating the performance of PG across different values of the behavior regularization coefficient $\alpha$. In practice, we sweep $\alpha \in \{50.0, 10.0, 1.0, 0.1, 0.01, 0.001\}$ in PG. Ablation results are provided in the Appendix H.

## 5   Conclusion

We introduced ***Prior Guidance***, a guided sampling method for diffusion planner-based offline RL that replaces the standard Gaussian prior with a learnable, behavior-regularized distribution. Unlike previous guided sampling methods, PG avoids inference-time overhead, reduces the risk of out-of-distribution actions, and enables stable behavior regularization in closed form. Empirically, it achieves strong performance across diverse tasks, especially in long-horizon domains.

**Limitations**   Despite its effectiveness, PG has several limitations. First, it introduces additional complexity through the training of both a latent value function and a learnable prior. Second, the architectural choices for the prior network have not been thoroughly investigated and warrant deeper exploration. Third, our evaluation is limited to offline settings and does not consider high-dimensional observations like images. Finally, the bijective mapping between latent noise and trajectory assumed by DDIM does not hold in practice due to limited discretization steps; this could be addressed by using flow-matching [35, 36], which we leave for future work.

# 6 Acknowledgements

This work was supported by the IITP(Institute of Information & Coummunications Technology Planning & Evaluation)-ITRC(Information Technology Research Center)(IITP-2025-RS-2024-00436857) grant funded by the Korea government(Ministry of Science and ICT) and by the IITP under the Artificial Intelligence Star Fellowship support program to nurture the best talents (IITP-2025-RS-2025-02304828) grant funded by the Korea government(MSIT).

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

# A   Related Works

**Learning Priors in RL**   Several prior works in reinforcement learning have utilized trainable prior distributions to achieve great success in multi-task learning [37] and skill learning [38, 39, 40, 41, 42]. [37] introduce a behavior prior framework that learns a distribution over trajectories to encode reusable movement and interaction patterns across tasks. [38] propose SPiRL, which leverages a learned skill prior to guide exploration and improve downstream task performance. [41] propose OPAL, which learns a latent prior over temporally extended primitive behaviors from offline data to improve downstream task performance in offline RL. [40] propose FIST, which learns a future-conditioned skill prior from offline data and adapts it for few-shot imitation of unseen long-horizon tasks. [39] propose SkiMo, which jointly learns a skill prior and a skill dynamics model from offline data to enable accurate long-horizon planning and improve downstream task performance. [42] propose NBDI, which learns a termination prior from a state-action novelty module to guide the execution of variable-length skills in downstream tasks.

While these methods learn priors over abstract skills or trajectories, they do not operate in the context of diffusion models. To the best of our knowledge, Prior Guidance (PG) is the first to replace the standard normal prior distribution of a diffusion model with a learnable prior in the context of reinforcement learning.

**Learning Priors in Diffusion Models**   Recent studies have explored methods to improve diffusion models by replacing the standard Gaussian prior with a more informative distribution. Notably, PriorGrad [43] proposes a data-dependent adaptive prior to accelerate training and improve sample quality in conditional generative tasks such as speech synthesis. While PriorGrad shares the motivation of improving the expressiveness and utility of the diffusion model's prior distribution, PriorGrad applies prior learning to the forward process in conditional generative models and leverages exact statistics from input conditions to define an instance-specific non-learned prior. In contrast, PG learns a parameterized prior distribution over the latent space of a behavior-cloned diffusion model, specifically targeting the reverse denoising process for high-return trajectory generation in offline RL.

**Behavior-Regularized Objectives in Offline RL**   To mitigate policy degradation caused by over-estimated value functions, a wide range of offline RL methods incorporate behavior-regularized objectives. [2] introduce BRAC, a unified framework that employs behavior regularization to stabilize policy learning. [26] propose Advantage-Weighted Regression (AWR), which formulates offline RL as supervised regression weighted by advantage estimates. [27] present AWAC, an actor-critic method that performs advantage-weighted supervised learning, enabling efficient offline pretraining and robust online fine-tuning. [44] propose OptiDICE, which optimizes a behavior-regularized objective over stationary distributions, avoiding the need for value bootstrapping or policy gradients. [4] introduce SQL and EQL, which leverage in-sample value regularization to address out-of-distribution (OOD) actions. [28] present SRPO, which directly regularizes the policy gradient using the score function of a pretrained diffusion model, thereby enabling behavior-regularized learning without sampling. [45] propose PORelDICE, which relaxes the positivity constraint in $f$-divergence regularization to better handle low-advantage samples. However, integrating behavior-regularized objectives into diffusion planners poses several challenges, such as requiring backpropagation through the iterative denoising process and making it difficult to handle a wide range of regularization functions in closed form.

**Diffusion Models in Offline RL**   Inspired by the success of diffusion models across a variety of domains [5, 6, 7, 8, 9, 10, 46], there has been growing interest in adapting diffusion models to offline RL. One line of research replaces conventional uni-modal Gaussian policies with expressive diffusion-based policies. For example, [20] propose Diffusion-QL, which combines conditional diffusion modeling with Q-learning guidance to improve policy expressiveness and regularization. [18] reinterpret IQL as an actor-critic algorithm and propose IDQL, which couples a generalized IQL critic with a diffusion-based behavior model. [33] introduce QGPO, which enables exact energy-guided diffusion sampling by aligning sampled trajectories with the energy landscape defined by a Q-function.

Another line of work leverages the temporal modeling capabilities of diffusion models to develop trajectory-level planners for long-horizon tasks. [11] introduce Diffuser, which formulates planning as conditional trajectory generation and enables flexible behavior synthesis via classifier-guided

sampling. [12] propose Decision Diffuser, which formulates decision-making as return-conditioned generative modeling without relying on value estimation. [13] present AdaptDiffuser, a self-evolving planner that iteratively improves its performance through reward-guided trajectory generation and model fine-tuning. [14] propose HDMI, a hierarchical diffusion model that generates subgoals conditioned on returns and performs trajectory denoising toward each subgoal. [15] propose Hierarchical Diffuser, a two-level architecture that combines jumpy subgoal generation with fine-grained trajectory refinement. Finally, [19] conduct a comprehensive empirical study of diffusion planners and introduce Diffusion Veteran, a simple baseline derived from key design principles such as sampling strategy, backbone architecture, and action generation scheme. However, existing guided sampling methods in diffusion planners suffer from several limitations—such as suboptimal sample selection, distributional shift, and high computational cost—highlighting the need for novel guidance approaches that can more effectively balance performance and efficiency.

**Concurrent Works**    A concurrent work, DSRL [47], introduces a method that steers a behavior-cloned diffusion policy by optimizing a learnable prior through reinforcement learning. While this idea of guiding generative models via a learned prior shares the same high-level goal as our approach, there exist both conceptual similarities and key methodological differences between DSRL and our PG framework.

Both DSRL and PG enhance decision quality by learning a prior distribution over the noise space $(\mathbf{x}_T \sim \mathcal{N}(\mathbf{0}, \mathbf{I}))$ that serves as input to the diffusion model's denoising process. In both methods, the prior is trained to maximize a value function defined over $\mathbf{x}_T$, which estimates the value of the corresponding denoised sample $\mathbf{x}_0$. This shared mechanism allows both methods to bias sampling toward high-value regions of the data or trajectory space.

However, there are fundamental differences. DSRL adopts a diffusion policy that directly maps noise to individual actions, whereas PG employs the diffusion planner that produces entire state trajectories and subsequently reconstructs executable actions via an inverse dynamics model. This trajectory-level formulation makes PG particularly effective in long-horizon or sparse-reward environments, where modeling temporal dependencies is critical for performance. Moreover, PG provides a theoretical foundation absent in DSRL. Specifically, we prove in Proposition 1 that, under sufficiently fine discretization, the optimization objective for the diffusion model can be equivalently reformulated as an optimization over the latent prior distribution. This equivalence arises from the bijective nature of the DDIM sampling process, which allows for an exact mapping between the data space and the latent prior. Finally, PG explicitly addresses the challenge of out-of-distribution (OOD) samples in offline reinforcement learning. While DSRL assumes the diffusion model generates only in-distribution actions and omits behavior constraints, our experiments (Figures 3 and 5) reveal that diffusion models trained on offline datasets can still produce OOD samples that harm performance. To mitigate this issue, PG introduces behavior regularization by imposing an $f$-divergence penalty between the learned prior and the standard normal distribution. This explicitly penalizes deviation from the behavior distribution and induces regularization at the prior level.

# B Proof of Proposition 1

**Proposition 1.** *Assume that DDIM sampling employs a sufficiently large number of discretization steps ensuring that the mapping $\mathbf{x_0} = g_{\mathbf{s}}(\mathbf{x}_T)$ is bijective. If the target trajectory density $\pi_{\psi}$ is parametrized by placing the prior $p_{\psi}(\mathbf{x}_T|\mathbf{s})$ while keeping the denoising process $g_{\mathbf{s}}$ fixed, the behavior-regularized objective* (2) *is equivalent to:*

$$\max_{\psi} \mathbb{E}_{\mathbf{s}\sim\mathcal{D},\mathbf{x}_T\sim p_{\psi}(\cdot|\mathbf{s})} \left[ V\left(g_{\mathbf{s}}\left(\mathbf{x}_T\right)\right) - \alpha f\left(\frac{p_{\psi}(\mathbf{x}_T|\mathbf{s})}{p(\mathbf{x}_T)}\right) \right]. \tag{3}$$

*Proof.* Since $\mathbf{x}_0 = g_s(\mathbf{x}_T)$ is bijective mapping between $\mathbf{x}_T$ and $\mathbf{x}_0$, we can apply the change-of-variables formula:

$$d\mathbf{x}_0 = d\mathbf{x}_T \cdot \left| \det\left(\frac{\partial g_{\mathbf{s}}(\mathbf{x}_T)}{\partial \mathbf{x}_T}\right) \right|.$$

Using this, we compute the conditional density $\tilde{\pi}_{\beta}(\mathbf{x}_0|\mathbf{s})$ as follows:

$$\tilde{\pi}_{\beta}(\mathbf{x}_0|\mathbf{s}) = \int_{\mathbf{x}_T} p(\mathbf{x}_T) \cdot \delta(\mathbf{x}_0 = g_{\mathbf{s}}(\mathbf{x}_T)) d\mathbf{x}_T$$

$$= \int_{\mathbf{x}_0} p(\mathbf{x}_T) \cdot \delta(\mathbf{x}_0 = g_{\mathbf{s}}(\mathbf{x}_T)) d\mathbf{x}_0 \cdot \left| \det\left(\frac{\partial g_{\mathbf{s}}(\mathbf{x}_T)}{\partial \mathbf{x}_T}\right) \right|^{-1}$$

$$= p(\mathbf{x}_T) \cdot \left| \det\left(\frac{\partial g_{\mathbf{s}}(\mathbf{x}_T)}{\partial \mathbf{x}_T}\right) \right|^{-1}.$$

Similarly, the density $\pi_{\psi}(\mathbf{x}_0|\mathbf{s})$ can be obtained in the same way:

$$\pi_{\psi}(\mathbf{x}_0|\mathbf{s}) = \int_{\mathbf{x}_T} p_{\psi}(\mathbf{x}_T|\mathbf{s}) \cdot \delta(\mathbf{x}_0 = g_{\mathbf{s}}(\mathbf{x}_T)) d\mathbf{x}_T$$

$$= \int_{\mathbf{x}_0} p_{\psi}(\mathbf{x}_T|\mathbf{s}) \cdot \delta(\mathbf{x}_0 = g_{\mathbf{s}}(\mathbf{x}_T)) d\mathbf{x}_0 \cdot \left| \det\left(\frac{\partial g_{\mathbf{s}}(\mathbf{x}_T)}{\partial \mathbf{x}_T}\right) \right|^{-1}$$

$$= p_{\psi}(\mathbf{x}_T|\mathbf{s}) \cdot \left| \det\left(\frac{\partial g_{\mathbf{s}}(\mathbf{x}_T)}{\partial \mathbf{x}_T}\right) \right|^{-1}.$$

Accordingly, we can reformulate the objective 2 as follows:

$$(2) = \max_{\psi} \mathbb{E}_{\mathbf{s}\sim D,\mathbf{x}_0\sim\pi_{\psi}(\cdot|\mathbf{s})} \left[ V(\mathbf{x}_0) - \alpha f\left(\frac{\pi_{\psi}(\mathbf{x}_0|\mathbf{s})}{\tilde{\pi}_{\beta}(\mathbf{x}_0|\mathbf{s})}\right) \right]$$

$$= \max_{\psi} \mathbb{E}_{\mathbf{s}\sim D,\mathbf{x}_T\sim p_{\psi}(\cdot|\mathbf{s})} \left[ V\left(g_{\mathbf{s}}(\mathbf{x}_T)\right) - \alpha f\left(\frac{p_{\psi}(\mathbf{x}_T|\mathbf{s}) \cdot \left| \det\left(\frac{\partial g_{\mathbf{s}}(\mathbf{x}_T)}{\partial \mathbf{x}_T}\right) \right|^{-1}}{p(\mathbf{x}_T) \cdot \left| \det\left(\frac{\partial g_{\mathbf{s}}(\mathbf{x}_T)}{\partial \mathbf{x}_T}\right) \right|^{-1}}\right) \right]$$

$$= \max_{\psi} \mathbb{E}_{\mathbf{s}\sim D,\mathbf{x}_T\sim p_{\psi}(\cdot|\mathbf{s})} \left[ V\left(g_{\mathbf{s}}(\mathbf{x}_T)\right) - \alpha f\left(\frac{p_{\psi}(\mathbf{x}_T|\mathbf{s})}{p(\mathbf{x}_T)}\right) \right]$$

$\square$

## C  Experimental Details

### C.1  Experiment Details of Figure 4

In Figure 4, we introduce a new baseline referred to as DQL (planner), which directly trains the diffusion model using a behavior-regularized objective (2). Specifically, it employs the diffusion behavior cloning loss (1) as a surrogate for behavior regularization, resulting in the following objective:

$$\max_{\psi} \mathbb{E}_{\mathbf{s} \sim D, \mathbf{x}_0 \sim \pi_{\psi}(\cdot|\mathbf{s})} \left[ V(\mathbf{x}_0) \right] - \alpha \mathcal{L}(\psi), \quad (6)$$

where $\mathcal{L}(\cdot)$ denotes the behavior cloning loss defined in Eq.(1). This approach can be seen as the planner version of Diffusion Q-Learning (DQL)[20], and we refer to it as DQL (planner) throughout the paper. The corresponding pseudo-code is provided in Algorithm 2. We measure the training time of DQL (planner) af-

---
**Algorithm 2** DQL (planner)

---
**Input:** Dataset $D$, Hyperparameter $\alpha$
**Require:** Inverse Dynamics $\epsilon_{\omega}$, Critic $V$
**Initialize:** Planner $\pi_{\psi}$

1: **// Training**
2: **for** each iteration **do**
3:      Sample $\mathbf{s} \sim D$ and $\mathbf{x}_T \sim p(\mathbf{x}_T)$
4:      Update planner $\pi_{\psi}$ using Eq. (6)
5: **end for**
6: **// Execution**
7: $\mathbf{s}^{(0)} = $ env.reset()
8: **for** environment step $\tau = 0, 1, \ldots$ **do**
9:      Sample $\mathbf{x}_0 \sim \pi_{\psi}(\cdot|\mathbf{s})$
10:     Predict action $\mathbf{a}^{(\tau)}$ using $\epsilon_{\omega}(\mathbf{x}_0)$
11:     $\mathbf{s}^{(\tau+1)} = $ env.step($\mathbf{a}^{(\tau)}$)
12: **end for**

---

ter aligning all other settings with those of Prior Guidance to ensure a fair comparison.

In Figure 4, we compare the training time of PG with and without backpropagation through the denoising process to specifically assess the efficiency of learning the latent value function. To this end, we exclude the time required to train the planner $g_{\mathbf{s}}$, inverse dynamics model $\epsilon_{\omega}$, and critic $V$, which are shared across both variants in Algorithm 1. Likewise, for DQL (planner), we report only the time taken to optimize the diffusion model for value maximization, excluding the time for training the inverse dynamics, critic, and the behavior cloning loss—which aligns with the planner training time in PG. This setup enables us to assess how much training time can be saved through latent value function learning.

### C.2  Experimental Settings

We evaluate Prior Guidance on the D4RL offline RL benchmark. For MuJoCo tasks, we report the average performance over 10 evaluation trajectories for each of 5 independently trained models. For Kitchen, AntMaze, and Maze2D tasks, we average over 100 evaluation trajectories for each of 5 independently trained models. In Prior Guidance, we adopt the same experimental settings for the planner $g_{\mathbf{s}}$, inverse dynamics model $\epsilon_{\omega}$, and critic $V$ as used in Diffusion Veteran (DV) [19]. We conducted all experiments using four NVIDIA RTX 4090 GPUs.

### C.3  Statistical Details

For all experiments, we report the mean and standard error of normalized scores computed over 5 training seeds. Additionally, in Figure 5, Figure 7b, and Table 2, we report the mean and standard error $\bar{\sigma}_{\text{avg}}$ of average normalized scores across multiple environments. The standard error $\bar{\sigma}_{\text{avg}}$ is computed using the error propagation formula based on the per-environment standard errors $\bar{\sigma}_i$, each estimated from 5 training seeds. Specifically,

$$\bar{\sigma}_{\text{avg}} = \sqrt{\sum_{i=1}^{n} \bar{\sigma}_i^2} \Big/ n, \quad \text{where } \bar{\sigma}_i = \frac{\sigma_i}{\sqrt{k}},$$

with $k = 5$ denoting the number of training seeds, $n$ the number of environments, and $\sigma_i$ the standard deviation for environment $i$.

# D   Diffusion Veteran (DV)

The pseudo-code for Diffusion Veteran (DV), which is identical to that of DV*, is provided in Algorithm 3.

---

**Algorithm 3** Diffusion Veteran (DV)

---

**Input:** Dataset $D$, Planning horizon $H$, Planning stride $m$, Candidate num $N$
**Initialize:** Planner $g_\mathbf{s}$, Inverse Dynamics $\epsilon_\omega$, Critic $V$

1: Compute discounted returns: $R^{(\tau)} = \sum_{h=0}^{\infty} \gamma^h r^{(\tau+h)}$  for every step $\tau$
2: // Training
3: **for** each iteration **do**
4:     Sample $\mathbf{s}^{(\tau)}, \mathbf{s}^{(\tau+m)}, \ldots, s^{(\tau+(H-1)m))}$ and $\mathbf{a}^{(\tau)}, R^{(\tau)}$ from $D$
5:     Update planner $g_\mathbf{s}$ using Eq. (1) with $\mathbf{s}^{(\tau)}$ as input, $\mathbf{s}^{(\tau)}, \ldots, \mathbf{s}^{(\tau+(H-1)m))}$ as target output
6:     Update inverse dynamics $\epsilon_\omega$ using Eq. (1) with $\mathbf{s}^{(\tau)}, \mathbf{s}^{(\tau+m)}$ as input, $\mathbf{a}^{(\tau)}$ as target output
7:     Update critic $V$ using MSE Loss with $\mathbf{s}^{(\tau)}, \ldots, \mathbf{s}^{(\tau+(H-1)m))}$ as input, $R^{(\tau)}$ as target output
8: **end for**
9: // Execution (MCSS)
10: $\mathbf{s}^{(0)}$ = env.reset()
11: **for** environment step $\tau$ = 0, 1, ... **do**
12:     Randomly generate $N$ candidate trajectories using $g_\mathbf{s}$ while fixing the first state as $\mathbf{s}^{(\tau)}$
13:     Select the trajectory with the highest estimated value $V$ among the $N$ candidates
14:     Generate $\mathbf{a}^{(\tau)}$ from the best trajectory using the inverse dynamics model $\epsilon_\omega$
15:     $\mathbf{s}^{(\tau+1)}$ = env.step($\mathbf{a}^{(\tau)}$)
16: **end for**

---

Full results corresponding to Figure 5, along with the comparison between the original implementation of DV and Prior Guidance, are provided in Table 3.

Table 3: Normalized scores on the D4RL benchmark. we compare Prior Guidance (PG) with the original implementation DV and DV* under different $N$s. We compute the mean and the standard error over 5 random seeds.

| | DV* | | | | | DV | PG |
|---|---|---|---|---|---|---|---|
| $N$ | 1 | 5 | 10 | 50 (default) | 500 | 50 | 1 |
| walker2d-medium-v2 | $70.8 \pm 2.2$ | $81.1 \pm 0.2$ | $80.6 \pm 0.3$ | $79.5 \pm 0.3$ | $79.5 \pm 0.7$ | $82.8 \pm 0.5$ | $82.3 \pm 0.2$ |
| walker2d-medium-replay-v2 | $40.3 \pm 1.8$ | $82.5 \pm 0.9$ | $84.1 \pm 0.4$ | $83.5 \pm 2.5$ | $85.9 \pm 0.4$ | $85.0 \pm 0.1$ | $83.7 \pm 1.0$ |
| walker2d-medium-expert-v2 | $96.1 \pm 2.1$ | $108.9 \pm 0.1$ | $109.0 \pm 0.1$ | $109.0 \pm 0.1$ | $108.9 \pm 0.2$ | $109.2 \pm 0.0$ | $109.4 \pm 0.1$ |
| hopper-medium-v2 | $47.2 \pm 0.9$ | $67.4 \pm 2.8$ | $70.1 \pm 3.5$ | $84.1 \pm 2.5$ | $91.5 \pm 1.7$ | $83.6 \pm 1.2$ | $97.5 \pm 0.6$ |
| hopper-medium-replay-v2 | $52.4 \pm 2.1$ | $91.2 \pm 0.2$ | $91.0 \pm 0.1$ | $91.3 \pm 0.1$ | $91.3 \pm 0.2$ | $91.9 \pm 0.0$ | $91.3 \pm 0.3$ |
| hopper-medium-expert-v2 | $76.2 \pm 5.9$ | $110.0 \pm 0.3$ | $110.2 \pm 0.5$ | $109.9 \pm 1.0$ | $110.6 \pm 0.9$ | $110.0 \pm 0.5$ | $110.4 \pm 0.1$ |
| halfcheetah-medium-v2 | $41.4 \pm 0.6$ | $47.5 \pm 0.1$ | $49.2 \pm 0.0$ | $50.9 \pm 0.1$ | $52.3 \pm 0.1$ | $50.4 \pm 0.0$ | $45.6 \pm 0.5$ |
| halfcheetah-medium-replay-v2 | $35.4 \pm 1.4$ | $43.2 \pm 0.6$ | $45.4 \pm 0.2$ | $46.4 \pm 0.2$ | $46.5 \pm 0.6$ | $45.8 \pm 0.1$ | $46.4 \pm 0.4$ |
| halfcheetah-medium-expert-v2 | $82.3 \pm 1.4$ | $91.8 \pm 0.2$ | $91.6 \pm 1.2$ | $92.3 \pm 0.6$ | $87.9 \pm 2.0$ | $92.7 \pm 0.3$ | $95.2 \pm 0.1$ |
| **Average** | 60.2 | 80.4 | 81.2 | 83.0 | 83.8 | 83.5 | **84.6** |
| kitchen-mixed-v0 | $52.4 \pm 0.7$ | $65.1 \pm 1.2$ | $70.3 \pm 1.7$ | $73.3 \pm 0.6$ | $67.5 \pm 3.0$ | $73.6 \pm 0.1$ | $74.6 \pm 0.4$ |
| kitchen-partial-v0 | $53.4 \pm 1.0$ | $77.7 \pm 2.0$ | $80.6 \pm 1.8$ | $76.2 \pm 4.3$ | $66.8 \pm 5.4$ | $94.0 \pm 0.3$ | $88.0 \pm 3.1$ |
| **Average** | 52.9 | 71.4 | 75.5 | 74.8 | 67.2 | **83.8** | 81.3 |
| antmaze-medium-play-v2 | $66.6 \pm 3.8$ | $86.4 \pm 1.4$ | $86.0 \pm 1.2$ | $80.8 \pm 1.3$ | $62.2 \pm 3.2$ | $89.0 \pm 1.6$ | $87.8 \pm 1.5$ |
| antmaze-medium-diverse-v2 | $63.6 \pm 2.0$ | $72.4 \pm 2.4$ | $71.8 \pm 4.0$ | $82.0 \pm 3.6$ | $84.6 \pm 1.6$ | $87.4 \pm 1.6$ | $87.3 \pm 1.7$ |
| antmaze-large-play-v2 | $74.8 \pm 1.3$ | $79.2 \pm 2.0$ | $80.6 \pm 3.4$ | $80.8 \pm 2.8$ | $81.0 \pm 2.0$ | $76.4 \pm 2.0$ | $82.4 \pm 2.1$ |
| antmaze-large-diverse-v2 | $78.8 \pm 2.6$ | $78.2 \pm 1.7$ | $77.6 \pm 3.0$ | $76.0 \pm 2.3$ | $72.0 \pm 1.4$ | $80.0 \pm 1.8$ | $76.0 \pm 1.9$ |
| **Average** | 71.0 | 79.1 | 79.0 | 79.9 | 75.0 | 83.2 | **83.4** |
| maze2d-umaze-v1 | $41.5 \pm 3.4$ | $97.1 \pm 3.5$ | $111.0 \pm 2.7$ | $133.8 \pm 1.9$ | $139.7 \pm 1.5$ | $136.6 \pm 1.3$ | $139.2 \pm 1.3$ |
| maze2d-medium-v1 | $23.0 \pm 3.5$ | $95.3 \pm 7.8$ | $123.8 \pm 5.9$ | $144.1 \pm 2.1$ | $151.2 \pm 2.6$ | $150.7 \pm 1.0$ | $159.5 \pm 0.8$ |
| maze2d-large-v1 | $54.6 \pm 5.1$ | $183.9 \pm 5.3$ | $190.8 \pm 5.0$ | $197.6 \pm 4.2$ | $201.8 \pm 5.5$ | $203.6 \pm 1.4$ | $195.2 \pm 2.3$ |
| **Average** | 39.7 | 125.4 | 141.9 | 158.5 | 164.2 | 163.6 | **164.6** |

# E   Full results of Table 1

Table 4: Normalized scores on the D4RL benchmark. We compute the mean and standard error over 5 random seeds.

| Dataset | | Gaussian policies | | | Diffusion policies | | | | | | Diffusion planners | | | | |
|---|---|---|---|---|---|---|---|---|---|---|---|---|---|---|---|
| | BC | CQL | IQL | SfBC | DQL | IDQL | QGPO | SRPO | Diffuser | AD | DD | HD | DV* | PG |
| Walker2d-M | 6.6 | 79.2 | 78.3 | 77.9±2.5 | 87.0±0.9 | 82.5 | 86.0±0.7 | 84.4±1.8 | 79.6±0.6 | 84.4±2.6 | 82.5±1.4 | 84.0±0.6 | 79.5±0.3 | 82.3±0.2 |
| Walker2d-M-R | 11.8 | 26.7 | 73.9 | 65.1±5.6 | 95.5±1.5 | 85.1 | 84.4±4.1 | 84.6±2.9 | 70.6±1.6 | 84.7±3.1 | 75.0±4.3 | 84.1±2.2 | 83.5±2.5 | 83.7±1.0 |
| Walker2d-M-E | 6.4 | 111.0 | 109.6 | 109.8±0.2 | 110.1±0.3 | 112.7 | 110.7±0.6 | 114.0±0.9 | 106.9±0.2 | 108.2±0.8 | 108.8±1.7 | 107.1±1.1 | 109.0±0.1 | 109.4±0.1 |
| Hopper-M | 29.0 | 58.0 | 66.3 | 57.1±4.1 | 90.5±4.6 | 65.4 | 98.0±2.6 | 95.5±0.8 | 74.3±1.4 | 96.6±2.7 | 79.3±3.6 | 99.3±0.3 | 84.1±2.5 | 97.5±0.6 |
| Hopper-M-R | 11.3 | 48.6 | 94.7 | 86.2±9.1 | 101.3±0.6 | 92.1 | 96.9±2.6 | 101.2±0.4 | 93.6±0.4 | 92.2±1.5 | 100.0±0.7 | 94.7±0.7 | 91.3±0.1 | 91.3±0.3 |
| Hopper-M-E | 111.9 | 98.7 | 91.5 | 108.6±2.1 | 111.1±1.3 | 108.6 | 108.0±2.5 | 100.1±5.7 | 103.3±1.3 | 111.6±2.0 | 111.8±1.8 | 115.3±1.1 | 109.9±1.0 | 110.4±0.1 |
| HalfCheetah-M | 36.1 | 44.4 | 47.4 | 45.9±2.2 | 51.1±0.5 | 51.0 | 54.1±0.4 | 60.4±0.3 | 42.8±0.3 | 44.2±0.6 | 49.1±1.0 | 46.7±0.2 | 50.9±0.1 | 45.6±0.5 |
| HalfCheetah-M-R | 38.4 | 46.2 | 44.2 | 37.1±1.7 | 47.8±0.3 | 45.9 | 47.6±1.4 | 51.4±1.4 | 37.7±0.5 | 38.3±0.9 | 39.3±4.1 | 38.1±0.7 | 46.4±0.2 | 46.4±0.4 |
| HalfCheetah-M-E | 35.8 | 62.4 | 86.7 | 92.6±0.5 | 96.8±0.3 | 95.9 | 93.5±0.3 | 92.2±1.2 | 88.9±0.3 | 89.6±0.8 | 90.6±1.3 | 92.5±0.3 | 92.3±0.6 | 95.2±0.1 |
| **Average** | 31.9 | 63.9 | 77.0 | 75.6 | 87.9 | 82.1 | 86.6 | 87.1 | 77.5 | 83.3 | 81.8 | 84.6 | 83.0 | 84.6 |
| Kitchen-M | 47.5 | 51.0 | 51.0 | 45.4±1.6 | 62.6±5.1 | 66.5 | – | – | 52.5±2.5 | 51.8±0.8 | 65.0±2.8 | 71.7±2.7 | 73.3±0.6 | 74.6±0.4 |
| Kitchen-P | 33.8 | 49.8 | 46.3 | 47.9±4.1 | 60.5±6.9 | 66.7 | – | – | 55.7±1.3 | 55.5±0.4 | 57.0±2.5 | 73.3±1.4 | 76.2±4.3 | 88.0±3.1 |
| **Average** | 40.7 | 50.4 | 48.7 | 46.7 | 61.6 | 66.6 | – | – | 54.1 | 53.7 | 61.0 | 72.5 | 74.8 | 81.3 |
| Antmaze-M-P | 0.0 | 14.9 | 71.2 | 81.3±2.6 | 76.6±10.8 | 84.5 | 83.6±4.4 | 80.7±2.9 | 6.7±5.7 | 12.0±7.5 | 8.0±4.6 | – | 80.8±1.3 | 87.8±1.5 |
| Antmaze-M-D | 0.0 | 15.8 | 70.0 | 82.0±3.1 | 78.6±10.3 | 84.8 | 83.8±3.5 | 75.0±5.0 | 2.0±1.6 | 6.0±3.3 | 4.0±2.8 | 88.7±8.1 | 82.0±3.6 | 87.3±1.7 |
| Antmaze-L-P | 0.0 | 53.7 | 39.6 | 59.3±14.3 | 46.4±8.3 | 63.5 | 66.6±9.8 | 53.6±5.1 | 17.3±1.9 | 5.3±3.4 | 0.0±0.0 | – | 80.8±2.8 | 82.4±2.1 |
| Antmaze-L-D | 0.0 | 61.2 | 47.5 | 45.5±6.6 | 56.6±7.6 | 67.9 | 64.8±5.5 | 53.6±2.6 | 27.3±2.4 | 8.7±2.5 | 0.0±0.0 | 83.6±5.8 | 76.0±2.3 | 76.0±1.9 |
| **Average** | 0.0 | 36.4 | 57.1 | 67.0 | 64.6 | 75.2 | 74.7 | 65.7 | 13.3 | 8.0 | 3.0 | – | 79.9 | 83.4 |
| Maze2D-U | 3.8 | 5.7 | 47.4 | 73.9±6.6 | – | 57.9 | – | – | 113.9±3.1 | 135.1±5.8 | – | 128.4±3.6 | 133.8±1.9 | 139.2±1.3 |
| Maze2D-M | 30.3 | 5.0 | 34.9 | 73.8±2.9 | – | 89.5 | – | – | 121.5±2.7 | 129.9±4.6 | – | 135.6±3.0 | 144.1±2.1 | 159.5±0.8 |
| Maze2D-L | 5 | 12.5 | 58.6 | 74.4±1.7 | – | 90.1 | – | – | 123.0±6.4 | 167.9±5.0 | – | 155.8±2.5 | 197.6±4.2 | 195.2±2.3 |
| **Average** | 13.0 | 7.7 | 47.0 | 74.0 | – | 79.2 | – | – | 119.5 | 144.3 | – | 139.9 | 158.5 | 164.6 |

# F  Hyperparameters

This section outlines the hyperparameters used to train Prior Guidance. The planner $g_s$, inverse dynamics model $\epsilon_\omega$, and critic $V$ all follow the same settings as those used in Diffusion Veteran. Detailed hyperparameter settings are provided in Table 5.

Table 5: Hyperparameter settings used for Prior Guidance on the D4RL benchmark.

| Settings | MuJoCo | Kitchen | Antmaze | Maze2d |
|---|---|---|---|---|
| Batch Size | 128 | 128 | 128 | 128 |
| State-Action Generation | Separate | Separate | Separate | Separate |
| Advantage Weighting | True | False | False | False |
| Inverse Dynamic | Diffusion | Diffusion | Diffusion | Diffusion |
| Time Credit Assignment | 0.997 | 0.997 | 1.0 | 1.0 |
| Planner Net. Backbone | Transformer | Transformer | Transformer | Transformer [48] |
| Transformer Hidden | 256 | 256 | 256 | 256 |
| Transformer Block | 2 | 2 | 2 | 8 |
| Planner Solver | DDIM | DDIM | DDIM | DDIM [25] |
| Planner Sampling Steps | 20 | 20 | 20 | 20 |
| Planner Training Steps | 1M | 1M | 1M | 1M |
| Planner Temperature | 1 | 1 | 1 | 1 |
| Planner Learning Rate | 2e-4 | 2e-4 | 2e-4 | 2e-4 |
| Planner Optimizer | AdamW | AdamW | AdamW | AdamW [49] |
| Planning Horizon | 4 | 32 | 32 | 40 |
| Planning Stride | 1 | 1 | 15 | 25 |
| Inverse Dynamics Net. Backbone | MLP | MLP | MLP | MLP |
| Inverse Dynamics Hidden | 256 | 256 | 256 | 256 |
| Inverse Dynamics Solver | DDPM | DDPM | DDPM | DDPM [5] |
| Inverse Dynamics Sampling Steps | 10 | 10 | 10 | 10 |
| Inverse Dynamics Training Steps | 1M | 1M | 1M | 1M |
| Policy Temperature | 0.5 | 0.5 | 0.5 | 0.5 |
| Policy Learning Rate | 3e-4 | 3e-4 | 3e-4 | 3e-4 |
| Policy Optimizer | AdamW | AdamW | AdamW | AdamW |
| Value Learning Rate | 3e-4 | 3e-4 | 3e-4 | 3e-4 |
| Value Net. Backbone | Transformer | Transformer | Transformer | Transformer |
| Value Optimizer | Adam | Adam | Adam | Adam [50] |
| Prior Network | GRU | GRU | GRU | GRU [30] |
| Prior Learning Rate | 3e-4 | 3e-4 | 3e-4 | 3e-4 |
| Prior Optimizer | AdamW | AdamW | AdamW | AdamW |
| Prior Hidden | 256 | 256 | 256 | 256 |
| Latent Value Learning Rate | 3e-4 | 3e-4 | 3e-4 | 3e-4 |
| Latent Value Net. Backbone | Transformer | Transformer | Transformer | Transformer |
| Latent Value Optimizer | Adam | Adam | Adam | Adam |
| Behavior Regularization | KL | KL | KL | KL |

In Table 15, we present ablation studies on the choice of behavior regularization function $f$. The corresponding hyperparameter $\alpha$ are listed in Table 6 and 7.

Table 6: $\alpha$ used for KL and Reverse KL-divergence on the D4RL benchmark.

| $\alpha$ | KL | Reverse KL |
|---|---|---|
| walker2d-medium-v2 | 0.1 | 0.1 |
| walker2d-medium-replay-v2 | 0.01 | 0.01 |
| walker2d-medium-expert-v2 | 0.01 | 0.01 |
| hopper-medium-v2 | 0.01 | 0.01 |
| hopper-medium-replay-v2 | 0.01 | 0.001 |
| hopper-medium-expert-v2 | 0.01 | 0.01 |
| halfcheetah-medium-v2 | 0.001 | 0.0001 |
| halfcheetah-medium-replay-v2 | 0.001 | 0.001 |
| halfcheetah-medium-expert-v2 | 0.01 | 0.01 |
| kitchen-mixed-v0 | 1.0 | 10.0 |
| kitchen-partial-v0 | 1.0 | 10.0 |
| antmaze-medium-play-v2 | 1.0 | 1.0 |
| antmaze-medium-diverse-v2 | 1.0 | 0.1 |
| antmaze-large-play-v2 | 50.0 | 1.0 |
| antmaze-large-diverse-v2 | 10.0 | 10.0 |
| maze2d-umaze-v1 | 1.0 | 10.0 |
| maze2d-medium-v1 | 0.1 | 1.0 |
| maze2d-large-v1 | 0.1 | 1.0 |

Table 7: $\alpha$ used for Pearson $\chi^2$-divergence on the D4RL benchmark.

| $\alpha$ | Pearson $\chi^2$ |
|---|---|
| walker2d-medium-v2 | 1e-4 |
| walker2d-medium-replay-v2 | 1e-7 |
| walker2d-medium-expert-v2 | 1e-6 |
| hopper-medium-v2 | 1e-8 |
| hopper-medium-replay-v2 | 1e-6 |
| hopper-medium-expert-v2 | 1e-5 |
| halfcheetah-medium-v2 | 1e-4 |
| halfcheetah-medium-replay-v2 | 1e-6 |
| halfcheetah-medium-expert-v2 | 1e-8 |
| kitchen-mixed-v0 | 1e-6 |
| kitchen-partial-v0 | 1e-7 |
| antmaze-medium-play-v2 | 1e-4 |
| antmaze-medium-diverse-v2 | 1e-2 |
| antmaze-large-play-v2 | 1e-4 |
| antmaze-large-diverse-v2 | 1e-2 |
| maze2d-umaze-v1 | 1e-7 |
| maze2d-medium-v1 | 1e-8 |
| maze2d-large-v1 | 1e-6 |

In Table 2, we report the average normalized scores on MuJoCo and AntMaze tasks when using a multi-modal Gaussian prior. The corresponding hyperparameter $\alpha$ are listed in Table 8.

Table 8: $\alpha$ used for multi-modal Gaussian prior on the D4RL benchmark.

| $\alpha$ | multi-modal |
|---|---|
| walker2d-medium-v2 | 0.01 |
| walker2d-medium-replay-v2 | 0.01 |
| walker2d-medium-expert-v2 | 0.001 |
| hopper-medium-v2 | 0.01 |
| hopper-medium-replay-v2 | 0.0001 |
| hopper-medium-expert-v2 | 0.01 |
| halfcheetah-medium-v2 | 0.0001 |
| halfcheetah-medium-replay-v2 | 0.001 |
| halfcheetah-medium-expert-v2 | 0.1 |
| antmaze-medium-play-v2 | 0.1 |
| antmaze-medium-diverse-v2 | 0.1 |
| antmaze-large-play-v2 | 0.1 |
| antmaze-large-diverse-v2 | 1.0 |

## G  Extensive Results

Full results of Figure 7b can be found in Table 9.

Table 9: Comparison of Prior Guidance with and without backpropagation through the denoising process (denoted as PG w/ Bp and PG w/o Bp, respectively) on the D4RL benchmark. We compute the mean and the standard error over 5 random seeds.

| | PG w/ Bp | PG w/o Bp |
|---|---|---|
| walker2d-medium-v2 | 81.3±0.2 | 82.3±0.2 |
| walker2d-medium-replay-v2 | 83.2±2.1 | 83.7±1.0 |
| walker2d-medium-expert-v2 | 109.6±0.2 | 109.4±0.1 |
| hopper-medium-v2 | 96.9±1.4 | 97.5±0.6 |
| hopper-medium-replay-v2 | 94.5±0.7 | 91.3±0.3 |
| hopper-medium-expert-v2 | 110.4±0.1 | 110.4±0.1 |
| halfcheetah-medium-v2 | 54.7±0.2 | 45.6± 0.5 |
| halfcheetah-medium-replay-v2 | 46.6±0.1 | 46.4±0.4 |
| halfcheetah-medium-expert-v2 | 95.1±0.2 | 95.2±0.1 |
| **Average** | 85.8 | 84.6 |
| kitchen-mixed-v0 | 74.3±0.6 | 74.6±0.4 |
| kitchen-partial-v0 | 87.6±4.8 | 88.0±3.1 |
| **Average** | 81.0 | 81.3 |
| antmaze-medium-play-v2 | 87.0±1.5 | 87.8±1.5 |
| antmaze-medium-diverse-v2 | 89.6±1.8 | 87.3±1.7 |
| antmaze-large-play-v2 | 82.2±2.4 | 82.4±2.1 |
| antmaze-large-diverse-v2 | 77.0±3.2 | 76.0±1.9 |
| **Average** | 84.0 | 83.4 |
| maze2d-umaze-v1 | 137.1±1.5 | 139.2±1.3 |
| maze2d-medium-v1 | 159.4±3.5 | 159.5±0.8 |
| maze2d-large-v1 | 196.2±2.6 | 195.2±2.3 |
| **Average** | 164.2 | 164.6 |

Full results of Table 2 can be found in Table 10.

Table 10: Normalized scores of PG using multi-modal Gaussian prior on MuJoCo and Antmaze tasks. We compute the mean and the standard error over 5 random seeds.

|  | multi-modal |
|---|---|
| walker2d-medium-v2 | 81.3±0.2 |
| walker2d-medium-replay-v2 | 82.5±0.2 |
| walker2d-medium-expert-v2 | 108.7±0.1 |
| hopper-medium-v2 | 88.9±2.3 |
| hopper-medium-replay-v2 | 96.2±0.0 |
| hopper-medium-expert-v2 | 110.5±0.1 |
| halfcheetah-medium-v2 | 44.3±2.1 |
| halfcheetah-medium-replay-v2 | 43.0±0.2 |
| halfcheetah-medium-expert-v2 | 84.9±0.5 |
| **Average** | 82.3 |
| antmaze-medium-play-v2 | 87.2±1.2 |
| antmaze-medium-diverse-v2 | 93.0±1.4 |
| antmaze-large-play-v2 | 83.4±1.4 |
| antmaze-large-diverse-v2 | 79.4±1.0 |
| **Average** | 85.8 |

## H Ablation Stuides

We conducted ablation studies on the behavior regularization coefficient $\alpha$ of PG across MuJoCo, Kitchen, AntMaze, and Maze2D tasks. The normalized scores of PG for different values of $\alpha$ are reported in Table 11, 12, 13 and 14.

Table 11: Ablation study of $\alpha$ in MuJoCo tasks. We compute the mean and the standard error over 5 random seeds.

| $\alpha$ | 0.001 | 0.01 | 0.1 | 1.0 | 10.0 |
|---|---|---|---|---|---|
| walker2d-medium-v2 | 77.2±0.6 | 81.4±0.3 | 82.3±0.2 | 74.5±0.8 | 70.8±2.2 |
| walker2d-medium-replay-v2 | 61.9±1.0 | 83.7±1.2 | 78.3±0.8 | 52.3±1.5 | 50.6±1.4 |
| walker2d-medium-expert-v2 | 109.4±0.1 | 109.1±0.1 | 108.6±0.2 | 108.4±0.1 | 108.5±0.2 |
| hopper-medium-v2 | 93.5±0.4 | 97.5±0.6 | 68.9±0.8 | 54.3±0.5 | 51.4±0.9 |
| hopper-medium-replay-v2 | 91.3±0.3 | 90.8±0.7 | 89.8±0.2 | 61.4±0.3 | 34.8±0.8 |
| hopper-medium-expert-v2 | 110.4±0.1 | 110.3±0.1 | 110.3±0.3 | 87.0±0.9 | 68.2±1.3 |
| halfcheetah-medium-v2 | 45.6±0.5 | 40.2±0.6 | 43.4±0.3 | 43.4±0.3 | 43.1±0.4 |
| halfcheetah-medium-replay-v2 | 46.4±0.4 | 45.0±0.2 | 44.1±0.4 | 43.0±0.5 | 39.0±0.5 |
| halfcheetah-medium-expert-v2 | 8.2±1.1 | 95.2±0.1 | 94.9±0.2 | 95.1±0.2 | 92.8±0.4 |

Table 12: Ablation study of $\alpha$ in Kitchen tasks. We compute the mean and the standard error over 5 random seeds.

| $\alpha$ | 0.1 | 1.0 | 10.0 |
|---|---|---|---|
| kitchen-mixed-v0 | 52.4±3.5 | 74.6±0.4 | 72.5±0.7 |
| kitchen-partial-v0 | 67.8±4.1 | 88.0±3.1 | 82.1±3.2 |

Table 13: Ablation study of $\alpha$ in Antmaze tasks. We compute the mean and the standard error over 5 random seeds.

| $\alpha$ | 0.1 | 1.0 | 5.0 | 10.0 | 50.0 |
|---|---|---|---|---|---|
| antmaze-medium-play-v2 | 47.0±4.2 | 87.8±1.5 | 83.8±1.6 | 84.0±1.3 | 80.2±1.5 |
| antmaze-medium-diverse-v2 | 86.0±1.4 | 87.3±1.7 | 80.3±1.8 | 72.1±2.5 | 68.4±2.8 |
| antmaze-large-play-v2 | 34.5±4.8 | 74.6±2.5 | 76.8±2.0 | 71.3±1.7 | 82.4±2.1 |
| antmaze-large-diverse-v2 | 21.8±3.2 | 70.4±1.8 | 69.7±1.8 | 76.0±1.9 | 68.5±2.1 |

Table 14: Ablation study of $\alpha$ in Maze2D tasks. We compute the mean and the standard error over 5 random seeds.

| $\alpha$ | 0.001 | 0.01 | 0.1 | 1.0 | 10.0 |
|---|---|---|---|---|---|
| maze2d-umaze-v1 | 113.3±2.5 | 91.9±3.7 | 126.3±1.7 | 139.2±1.3 | 122.2±2.1 |
| maze2d-medium-v1 | 155.6±0.8 | 121.1±4.8 | 159.5±0.8 | 148.4±1.7 | 152.1±1.2 |
| maze2d-large-v1 | 143.4±5.1 | 151.3±4.7 | 195.2±2.3 | 185.0±5.3 | 183.9±4.8 |

We also conducted ablation studies on various behavior regularization functions $f$, as shown in Table 15.

Table 15: Comparison of normalized scores of Prior Guidance across four behavior regularization functions—KL, Reverse KL, and Pearson $\chi^2$—on the D4RL benchmark. We compute the mean and the standard error over 5 random seeds.

| $f$ | KL | Reverse KL | Pearson-$\mathcal{X}^2$ |
|---|---|---|---|
| walker2d-medium-v2 | 82.3±0.2 | 82.0±0.3 | 78.6±1.4 |
| walker2d-medium-replay-v2 | 83.7±1.0 | 83.7±0.4 | 82.4±0.3 |
| walker2d-medium-expert-v2 | 109.4±0.1 | 109.1±0.0 | 103.5±0.2 |
| hopper-medium-v2 | 97.5±0.6 | 96.5±0.9 | 97.0±0.1 |
| hopper-medium-replay-v2 | 91.3±0.3 | 93.5±0.0 | 92.4±0.0 |
| hopper-medium-expert-v2 | 110.4±0.1 | 110.3±0.0 | 111.5±0.0 |
| halfcheetah-medium-v2 | 45.6± 0.5 | 48.8±0.5 | 43.5±0.1 |
| halfcheetah-medium-replay-v2 | 46.4±0.4 | 46.0±0.1 | 44.9±0.1 |
| halfcheetah-medium-expert-v2 | 95.2±0.1 | 95.4±0.1 | 92.5±0.1 |
| **Average** | 84.6 | 85.0 | 82.9 |
| kitchen-mixed-v0 | 74.6±0.4 | 73.3±0.3 | 71.2±0.3 |
| kitchen-partial-v0 | 88.0±3.1 | 85.3±1.0 | 75.6±3.7 |
| **Average** | 81.3 | 79.3 | 73.4 |
| antmaze-medium-play-v2 | 87.8±1.5 | 87.6±2.2 | 76.3±2.5 |
| antmaze-medium-diverse-v2 | 87.3±1.7 | 86.3±2.0 | 80.0±2.3 |
| antmaze-large-play-v2 | 82.4±2.1 | 79.7±1.8 | 85.2±1.5 |
| antmaze-large-diverse-v2 | 76.0 ± 1.9 | 77.0±1.9 | 70.9±2.2 |
| **Average** | 83.4 | 82.7 | 78.1 |
| maze2d-umaze-v1 | 139.2±1.3 | 136.2±2.1 | 132.0±2.2 |
| maze2d-medium-v1 | 159.5±0.8 | 150.4±1.6 | 170.8±1.0 |
| maze2d-large-v1 | 195.2±2.3 | 186.6±2.3 | 193.2±2.0 |
| **Average** | 164.6 | 157.7 | 165.3 |

# I   Why PG underperform on standard MuJoCo tasks?

Table 16: Performance comparison across standard MuJoCo tasks.

|  | DQL | DV* | PG (planner) | PG (policy) |
|---|---|---|---|---|
| walker2d-medium-v2 | **87.0** | 79.5 | 82.3 | $84.5 \pm 0.26$ |
| walker2d-medium-replay-v2 | 95.5 | 83.5 | 83.7 | **96.4** $\pm 2.49$ |
| walker2d-medium-expert-v2 | **110.1** | 109.0 | 109.4 | $109.8 \pm 0.23$ |
| hopper-medium-v2 | 90.5 | 84.1 | 97.5 | **98.4** $\pm 1.01$ |
| hopper-medium-replay-v2 | **101.3** | 91.3 | 91.3 | $100.5 \pm 1.27$ |
| hopper-medium-expert-v2 | 111.1 | 109.9 | 110.4 | **111.4** $\pm 0.57$ |
| halfcheetah-medium-v2 | 51.1 | 50.9 | 45.6 | **54.7** $\pm 0.42$ |
| halfcheetah-medium-replay-v2 | 47.8 | 46.4 | 46.4 | **48.6** $\pm 0.34$ |
| halfcheetah-medium-expert-v2 | **96.8** | 92.3 | 95.2 | $95.5 \pm 0.26$ |
| Average | 87.9 | 83.0 | 84.6 | **88.9** |

PG is a diffusion planner-based method designed for long-horizon planning, where capturing temporally extended dependencies is essential. However, MuJoCo locomotion tasks involve short-horizon, dense-reward control focused on making agents run faster, which does not require such lookahead capabilities [19]. As a result, diffusion planner-based methods, including PG, often underperform compared to diffusion policy-based approaches in these settings, as shown in Table 1. While the additional structure and trajectory-level modeling of planners benefit challenging long-horizon domains such as AntMaze, Kitchen, or Maze2D, they provide little advantage and can even be slightly detrimental in simpler environments where policies can already achieve near-optimal performance.

To examine whether the observed performance gap in MuJoCo tasks from the use of the diffusion planner rather than a limitation of the PG framework itself, we conduct an additional experiment by applying PG to a diffusion policy. Specifically, we replaced the standard Gaussian prior of a behavior-cloned diffusion policy with a learnable, behavior-regularized prior, following the same training procedure used in PG.

Table 16 shows that replacing the diffusion planner in PG with a diffusion policy (PG policy) consistently improved performance across MuJoCo tasks, raising the average score from 84.6 (PG planner) to 88.9 and surpassing the DQL baseline score of 87.9. This demonstrates that PG remains effective even in short-horizon environments, suggesting that the lower performance of PG in MuJoCo tasks is due to the nature of trajectory-level planning rather than a limitation of the method itself. The results for PG (policy) are averaged over 3 random seeds and reported as the mean $\pm$ standard error.

