# OpenReview forum: "Prior-Guided Diffusion Planning for Offline Reinforcement Learning"
_NeurIPS.cc/2025/Conference — NeurIPS 2025 poster_

### Official Review · Reviewer_rn8g · 2025-06-29

**Clarity:** 4
**Significance:** 3
**Originality:** 3
**Rating:** 4
**Confidence:** 3

**Summary:**

This paper introduces Prior Guidance (PG) an offline RL approach to guide the diffusion-based planner. The authors argue that current diffusion-based guidance methods including Classifier Guidance and Classifier-Free Guidance and Monte Carlo Sample Selection (MCSS) suffer from limitations such as computational inefficiency, overfitting to out-of-distribution (OOD) actions etc.

PG proposes to replace the standard Gaussian prior in the difussion model with a learned, state-conditioned Gaussian prior, optimizing using a behavior-regularized objective. To avoid backpropagation through the diffusion process, the authors introduce a latent value function that operates directly in the prior space, enabling efficient training and inference.

PG is evaluated empirically on long-horizon planning tasks such as D4RL, Mujoco control suit and Maze2D-U.

**Questions:**

- Can you clarify which specific Kitchen tasks were used? Is the reported score averaged over several tasks (e.g., kitchen-mixed-v0, kitchen-partial-v0, etc.) or based on a single one?
- Why does PG underperform on standard MuJoCo tasks? These tasks are supposed to be easy and saturated within the RL domain. Do you attribute this to horizon length, simplicity of dynamics, or limitations of your prior formulation?
- How is the inverse dynamics model trained and evaluated?

**Ethical Concerns:**

["NO or VERY MINOR ethics concerns only"]

**Limitations:**

- Domain sensitivity: PG performs well on temporally extended tasks (kitchen, Maze2D-U) but fails to match baseline in standard control environments, suggesting its benefits are limited to certain settings.
- Because the denoising model is frozen, PG may not adapt well to high-return regions discovered during training - If I understood correctly, PG freezes the diffusion model after behavior cloning. So only the prior is trained to bias generation towards higher-value trajectories. But now suppose the value function discovers a new region in latent space where higher returns can be achieved. Now if we want to shift the prior to sample more from this region but the decoder is unchanged, it may not accurately reconstruct those high-return trajectories.
- Restricting the prior to a Gaussian form limits expressiveness

**Quality:**

3

**Strengths And Weaknesses:**

## Strengths
- Reformulating the guidance objective in the latent space and optimizing the prior directly is theoretically sound and overall well-motivated.
- PG avoids the high inference-time cost of MCSS and eliminates backpropagation through denoising network.
- PG enables tractable, analytically stable regularization by performing directly in the prior space
- Good performance across some of the baselines including Kitchen add Maze2D-U, which require multi-step planning and skill composition.
- Overall paper is well-written and easy to follow.
- The paper provides informative comparisons and ablations between PG and without denoising backpropagation as wel as existing methods like MCSS and Diffusion Q-learning.

## Weaknesses
- I have some concerns with regards to the empirical results. In Mujoco control environments like Walker2d, HalfCheetah and Hopper PG is consistently outperformed by existing baselines. This raises questions about its generality beyond long-horizon planning.
- It was also not clear to me what constitutes the Kitchen benchmark, the paper keeps refering to "Kitchen" without explaining that this is the FrankaKitchen environment from D4RL? For readers unfamiliar with it, the task complexity and relevance remain unclear (and whether the single reported results is the average across multiple kitchen tasks). It might be more helpful to see performance of individual tasks since this method has shortcomings in some of these mujoco environments.
- Theoretical justification hinges on DDIM being bijective which in parctice it may not hold (because it's an approximate ODE solver)
- PG uses a relatively simple Gaussian prior which may not be sufficient in stochastic behavior setting

Overall, I like the ideas in this paper. I would be open to raising my score if the authors can convincingly explain the reasons behind the weaker performance on standard control tasks and provide assurance that these results do not reflect a fundamental flaw in the proposed method.

---

> ### Author Rebuttal · Authors · 2025-07-31
>
> **Dear Reviewer rn8g,**
>
> We appreciate your thoughtful and detailed review. We hope that the explanations provided below will address your questions and resolve any concerns..
>
> ---
>
> **W3. Theoretical justification hinges on DDIM being bijective which in parctice it may not hold (because it's an approximate ODE solver)**
>
> We acknowledge that exact bijectivity is not guaranteed in practice under standard DDIM sampling with finite discretization steps. This is a known limitation of DDIM inversion [1, 2], which we have also noted in the Limitations section. To fully ensure bijectivity, one could employ an ODE-based generative model or a flow-matching framework, both of which offer theoretical invertibility.
>
> Nevertheless, our empirical results indicate that the bijective assumption holds approximately in practice and is sufficient for effective guidance. As shown in Figure 6, noise samples drawn from the learned prior consistently denoise into high‑value trajectories, indicating that the mapping from latent space to trajectory space is preserved well enough to enable strong performance. Consequently, as reported in Table 1, PG achieves state‑of‑the‑art results across diverse benchmarks, including long‑horizon planning tasks such as AntMaze, Kitchen, and Maze2D.
>
> These results suggest that while exact bijectivity is not guaranteed, the approximate invertibility of DDIM is sufficient in practice for guiding diffusion models via learned priors.
>
> ---
>
> **W4. PG uses a relatively simple Gaussian prior which may not be sufficient in stochastic behavior setting.**
>
> We appreciate the reviewer’s insightful comment regarding the potential limitations of using a simple Gaussian prior in stochastic behavior settings. We agree that a more expressive prior model, such as a Gaussian mixture model (GMM), could be beneficial in scenarios where multiple distinct trajectories achieve similarly high returns, even under a deterministic optimal policy. To investigate this, we evaluated PG with a multi‑modal prior in our experiments (Table 2) to assess whether such an expressiveness would improve performance, and found that the improvement was not substantial. Nonetheless, PG can be easily extended to incorporate a more expressive prior, and our framework is fully compatible with such an extension if needed.
>
> ---
>
> **W2, Q1. Can you clarify which specific Kitchen tasks were used? Is the reported score averaged over several tasks (e.g., kitchen-mixed-v0, kitchen-partial-v0, etc.) or based on a single one?**
>
> We thank the reviewer for the helpful question and apologize for the lack of clarity. We clarify that the **"Kitchen" domain refers to the FrankaKitchen environment from the D4RL benchmark suite** [1]. In Table 1, we report performance on two standard Kitchen tasks from the D4RL benchmark suite: kitchen-mixed-v0 (denoted as Kitchen-M) and kitchen-partial-v0 (denoted as Kitchen-P). The “Kitchen” score, as reported in Figure 5, and Figure 7(b), refers to the average of the normalized scores across these two tasks. We will add detailed clarification on Section 4.1 in the final version.
>
> ---
>
> **W1, Q2. Why does PG underperform on standard MuJoCo tasks? These tasks are supposed to be easy and saturated within the RL domain. Do you attribute this to horizon length, simplicity of dynamics, or limitations of your prior formulation?**
>
> We appreciate the reviewer’s insightful comment on PG’s performance in standard MuJoCo tasks. PG is a diffusion planner-based method designed for long-horizon planning, where capturing temporally extended dependencies is essential. However, MuJoCo locomotion tasks involve short-horizon, dense-reward control focused on making agents run faster, which does not require such lookahead capabilities [2]. As a result, diffusion planner-based methods, including PG, often underperform compared to diffusion policy-based approaches in these settings, as shown in Table 1 of our paper. While the additional structure and trajectory-level modeling of planners benefit challenging long-horizon domains such as AntMaze, Kitchen, or Maze2D, they provide little advantage and can even be slightly detrimental in simpler environments where policies can already achieve near-optimal performance.
>
> |                   | DQL        | DV*       | PG (planner) | PG (policy)        |
> |:-----------:|:--------------:|:-------------:|:----------------:|:----------------------:|
> | Walker2d-M  | **87.0**       | 79.5          | 82.3              | 84.5 $\pm$ 0.26         |
> | Walker2d-M-R| 95.5           | 83.5          | 83.7              | **96.4 $\pm$ 2.49**      |
> | Walker2d-M-E| **110.1**      | 109.0         | 109.4             | 109.8 $\pm$ 0.23        |
> | Hopper-M    | 90.5           | 84.1          | 97.5              | **98.4 $\pm$ 1.01**     |
> | Hopper-M-R  | **101.3**      | 91.3          | 91.3              | 100.5 $\pm$ 1.27        |
> | Hopper-M-E  | 111.1          | 109.9         | 110.4             | **111.4 $\pm$ 0.57**    |
> | HalfCheetah-M| 51.1          | 50.9          | 45.6              | **54.7 $\pm$ 0.42**     |
> | HalfCheetah-M-R| 47.8        | 46.4          | 46.4              | **48.6 $\pm$ 0.34**     |
> | HalfCheetah-M-E| **96.8**    | 92.3          | 95.2              | 95.5 $\pm$ 0.26         |
> | Average | 87.9       | 83.0          | 84.6              | **88.9**                |
>
> To examine whether the observed performance gap in MuJoCo tasks stems from the use of a diffusion planner rather than a limitation of the PG framework itself, we conduct an additional experiment by applying PG to a diffusion policy. Specifically, we replaced the standard Gaussian prior of a behavior-cloned diffusion policy with a learnable, behavior-regularized prior, following the same training procedure used in PG.
>
> The above table shows that replacing the diffusion planner in PG with a diffusion policy (PG policy) consistently improved performance across MuJoCo tasks, raising the average score from 84.6 (PG planner) to 88.9 and surpassing the DQL baseline score of 87.9. This demonstrates that PG remains effective even in short-horizon environments, suggesting that the lower performance of PG in MuJoCo tasks is due to the nature of trajectory-level planning rather than a limitation of the method itself. The results for PG (policy) are averaged over 3 random seeds and reported as the mean $\pm$ standard error.
>
> We will include these results to further clarify that the PG framework is effective across both policy- and planner-based approaches.
>
> ---
>
> **Q3. How is the inverse dynamics model trained and evaluated?**
>
> We apologize for the lack of explanation regarding the inverse dynamics model. In Prior Guidance (PG), the inverse dynamics model $\epsilon_w$ is trained following the setup established in Diffusion Veteran (DV), which PG builds upon.  The model maps a pair of temporally spaced states $(\mathbf{s}^{(\tau)}, \mathbf{s}^{(\tau+m)})$ to the action $\mathbf{a}^{(\tau)}$ , where $\tau$ denotes the current environment step, and $m$ is the planning stride. Then $\epsilon_w$ is used to recover executable actions from denoised state trajectories $\mathbf{x}_0 = \left[ \mathbf{s}^{(\tau)}, \mathbf{s}^{(\tau + m)}, \dots, \mathbf{s}^{(\tau + (H-1)m)} \right]$ generated by the diffusion planner.
>
> **Training:**
>
> - The inverse model is trained via supervised regression from the dataset $\mathcal{D}$. The objective is to minimize the following loss:
>
>     $$
>     \mathbb{E}_{\mathbf{s}^{(\tau)}, \mathbf{s}^{(\tau + m)}, \mathbf{a}^{(\tau)},\epsilon,t} \left[\left\Vert \epsilon - \epsilon_w \left(\sqrt{\bar{\alpha}_t}  \mathbf{a}^{(\tau)} + \sqrt{1 - \bar{\alpha}_t}\epsilon, t, s^{(\tau)},s^{(\tau+m)}\right) \right\Vert^2\right]
>     $$
>
> **Inference:**
>
> - At test time, after PG generates a full predicted state trajectory  $\mathbf{x}_0 = \left[ \mathbf{s}^{(\tau)}, \mathbf{s}^{(\tau + m)}, \dots, \mathbf{s}^{(\tau + (H-1)m)} \right]$, and the action $\mathbf{a}^{(\tau)}$  is computed by applying the inverse dynamics model $\epsilon_w$ to the first two states $\left(\mathbf{s}^{(\tau)}, \mathbf{s}^{(\tau + m)}\right)$
> - This action is then executed in the environment, and the process repeats at the next timestep.
>
> We will include this detailed description of the inverse dynamics model in the next revision.
>
> ---
>
> **References**
>
> [1] Fu, Justin, et al. "D4rl: Datasets for deep data-driven reinforcement learning." *arXiv preprint arXiv:2004.07219* (2020).
>
> [2] Lu, Haofei, et al. "What makes a good diffusion planner for decision making?." *arXiv preprint arXiv:2503.00535* (2025).

---

### Official Review · Reviewer_6FdX · 2025-07-02

**Clarity:** 3
**Significance:** 2
**Originality:** 2
**Rating:** 4
**Confidence:** 4

**Summary:**

This paper introduces Prior Guidance (PG), a novel method for improving diffusion-based planners in offline reinforcement learning (RL).
the key idea is to learn a prior conditional on the state instead the gaussian prior in order to steer the diffusion samples to high return regions. At training, the author propose to learn a latent value function (value function that depends on the noise sampled from the prior) to avoid backpropagate gradient through the denoising steps. at inference time, we need just sample from the learned prior which save runtime comparing to other guidance method (classifer guidance or Monte Carlo sample selection)

**Questions:**

see weakness above

**Ethical Concerns:**

["NO or VERY MINOR ethics concerns only"]

**Final Justification:**

I maintain my score. My main concern is still the limited evaluation to D4RL where we see very moderate boost of performance.

**Limitations:**

yes

**Paper Formatting Concerns:**

no issues

**Quality:**

3

**Strengths And Weaknesses:**

Strengths
- Computational Efficiency: Reduces inference time significantly by avoiding sampling multiple candidate trajectories (unlike Monte Carlo methods).
- Avoids backpropagation through the denoising process by using a latent-space critic.
- Closed-form Regularization: Behavior regularization is done in the latent space where densities are tractable (Gaussian), allowing for closed-form solutions and improved stability.
- The method builds on existing diffusion planners with minimal changes: just replace the prior and train an auxiliary value function.

Weakness:

- Additional Complexity: comparing to MCSS, the proposed method requires training and tuning of a latent-space value function and a learnable prior, adding architectural and training complexity.
- in order to get a tractable regularization term, the author assumes bijectivity of the denoising scheme, which is not really satisfied.
- Empirically speaking, while the proposed seem to work well, the performance boost is not too important. I think the performance is quite saturated in D4RL Benchmark. it would be important in include more challenging domains (e.g long-horizon tasks in ogbench, humanoid-large-maze ?)

---

> ### Author Rebuttal · Authors · 2025-07-31
>
> **Dear Reviewer 6FdX,**
>
> Thank you for your valuable comments. We hope that the following responses will answer your questions and address the concerns you have raised.
>
> ---
>
> **W1. Additional Complexity: comparing to MCSS, the proposed method requires training and tuning of a latent-space value function and a learnable prior, adding architectural and training complexity.**
>
> We acknowledge that Prior Guidance (PG) introduces additional networks, which we have also noted in the Limitations section, and that this indeed increase architectural and training complexity relative to MCSS. However, we highlight that this added complexity is both minimal and reasonable given the substantial gains in performance and efficiency.
>
> 1. **Lower Training Cost**
>
>     The introduced components are lightweight and modular. The prior is parameterized by a simple GRU and the latent value function is a shallow MLP, both adding minor overhead compared to the diffusion planner backbone. Moreover, these networks are trained separately from the planner without backpropagation through the diffusion model, keeping the overall computational cost manageable (see Figure 4, left).
>
> 2. **Avoidance of Costly Sampling**
>
>     MCSS requires sampling and scoring a large number of candidate trajectories at inference time, which is both computationally expensive and difficult to scale, especially in long-horizon or high-dimensional settings. PG eliminates this bottleneck by generating a single trajectory, yielding up to a 500× reduction in inference cost (see Figure 4, right).
>
> 3. **Performance Validates Overhead**
>
>     Despite the small additional training components, PG consistently outperforms MCSS across most long-horizon benchmarks (Table 1, Figure 5), demonstrating that the modest increase in training complexity is outweighed by substantial gains in both efficiency and performance.
>
> ---
>
> **W2. in order to get a tractable regularization term, the author assumes bijectivity of the denoising scheme, which is not really satisfied.**
>
> We acknowledge that exact bijectivity is not guaranteed in practice under standard DDIM sampling with finite discretization steps. This is a known limitation of DDIM inversion [1, 2], which we have also noted in the Limitations section. To fully ensure bijectivity, one could employ an ODE-based generative model or a flow-matching framework, both of which offer theoretical invertibility.
>
> Nevertheless, our empirical results indicate that the bijective assumption holds approximately in practice and is sufficient for effective guidance. As shown in Figure 6, noise samples drawn from the learned prior consistently denoise into high‑value trajectories, indicating that the mapping from latent space to trajectory space is preserved well enough to enable strong performance. Consequently, as reported in Table 1, PG achieves state‑of‑the‑art results across diverse benchmarks, including long‑horizon planning tasks such as AntMaze, Kitchen, and Maze2D.
>
> These results suggest that while exact bijectivity is not guaranteed, the approximate invertibility of DDIM is sufficient in practice for guiding diffusion models via learned priors.
>
> ---
>
> **W3. Empirically speaking, while the proposed seem to work well, the performance boost is not too important. I think the performance is quite saturated in D4RL Benchmark. it would be important in include more challenging domains (e.g long-horizon tasks in ogbench, humanoid-large-maze ?)**
>
> We thank the reviewer for this valuable suggestion. We agree that D4RL has been extensively studied and that evaluating on more challenging domains, such as long‑horizon tasks in OGBench or humanoid‑large‑maze, would further validate the generality of our approach. However, given the limited rebuttal period, conducting such large‑scale experiments is unfortunately infeasible. We believe that PG clearly outperforms other baselines in long-horizon planning tasks within D4RL, while also reducing inference time substantially compared to the previous state-of-the-art method DV, constitutes a meaningful contribution. We thank the reviewer again for this suggestion and plan to extend our evaluation to more challenging environments in future work.
>
> ---
>
> **References**
>
> [1] Song, Jiaming, Chenlin Meng, and Stefano Ermon. "Denoising diffusion implicit models." *arXiv preprint arXiv:2010.02502* (2020).
>
> [2] Dhariwal, Prafulla, and Alexander Nichol. "Diffusion models beat gans on image synthesis." Advances in neural information processing systems 34 (2021): 8780-8794.

---

> ### Comment · Reviewer_6FdX · 2025-08-05
>
> Thanks for the rebuttal. I maintain my score.
> My main concern is still the limited evaluation to D4RL where we see very moderate boost of performance.

---

### Official Review · Reviewer_J4nL · 2025-07-02

**Clarity:** 3
**Significance:** 3
**Originality:** 3
**Rating:** 4
**Confidence:** 3

**Summary:**

This paper introduces a novel method, Prior Guidance (PG), into the standard diffusion planning framework. Instead of sampling from a fixed Gaussian prior, the method samples from a learnable prior, which allows the policy to be optimized via a behavior-regularized objective. Experimental results demonstrate that PG performs well across multiple tasks, particularly on long-horizon problems. It also outperforms standard Monte Carlo Sample Selection (MCSS) methods in both performance and computational efficiency.

**Questions:**

1. Why does the proposed PG method not perform well on some datasets? For example, in Table 1, PG does not outperform some diffusion-based methods such as DQL in MuJoCo tasks. Does the method only work well on long-horizon tasks?

2. How does the method achieve good performance on long-horizon tasks, consistently outperforming MCSS? It seems that prior guidance alone does not provide this capability. Does the improvement come from Diffusion Veteran (DV)?

3. It is unclear whether the method modifies the training of the base Diffusion Veteran. Is the proposed prior guidance applied only at the sampling stage?

**Ethical Concerns:**

["NO or VERY MINOR ethics concerns only"]

**Final Justification:**

I thank the reviewer for the detailed responses in the rebuttal. The results seem promising; however, the lack of an ablation study on the effect of dropping the Diffusion Veteran (DV) makes the contribution of the proposed method to the overall performance unclear. Nevertheless, the method itself appears promising, so I will maintain my current evaluation.

**Limitations:**

The limitations of this work are discussed in the paper.

**Paper Formatting Concerns:**

There are no major formatting issues in the paper.

**Quality:**

3

**Strengths And Weaknesses:**

Strengths:

1. The idea of introducing a learnable prior into the sampling process of diffusion planning is quite novel.

2. Theoretical analysis is provided to justify the training objective of prior guidance and its validity.

3. The paper presents comprehensive experimental results, offering a convincing evaluation of the proposed method.

4. Concrete and detailed implementation details are provided in the appendix.

Weaknesses:

1. Although the proposed PG method performs well on some long-horizon tasks, it does not show better performance on MuJoCo simulation tasks compared to standard diffusion-based policies such as Diffusion-QL.

2. It is unclear how much of the performance gain comes from the incorporation of Diffusion Veteran (DV). The authors are encouraged to include an ablation study using the standard Diffuser as the base model to better isolate the contribution of PG.

---

> ### Author Rebuttal · Authors · 2025-07-31
>
> **Dear Reviewer J4nL,**
>
> We appreciate your valuable comments. We hope that our responses below will clarify the points raised and resolve any concerns.
>
> ---
>
> **W2. It is unclear how much of the performance gain comes from the incorporation of Diffusion Veteran (DV). The authors are encouraged to include an ablation study using the standard Diffuser as the base model to better isolate the contribution of PG.**
>
> We thank the reviewer for highlighting the importance of isolating the contribution of PG from that of Diffusion Veteran (DV). Due to the limited time available during the rebuttal period, we were unable to conduct an ablation using the standard Diffuser as the base model. Instead, we performed a related experiment by applying PG to a diffusion policy that uses the same diffusion model architecture as DQL, allowing us to assess whether PG provides meaningful performance gains even without incorporating DV. As detailed in our response to Q1, applying PG by substituting the standard Gaussian prior of a behavior-cloned diffusion policy with a learned, behavior-regularized prior achieved an average score of 88.9, outperforming other diffusion-based policies on MuJoCo tasks. These results demonstrate that PG provides a meaningful performance gain even without DV, indicating that its benefits are not tied to a specific base model.
>
> ---
>
> **W1, Q1. Why does the proposed PG method not perform well on some datasets? For example, in Table 1, PG does not outperform some diffusion-based methods such as DQL in MuJoCo tasks. Does the method only work well on long-horizon tasks?**
>
> We appreciate the reviewer’s insightful comment on PG’s performance in MuJoCo tasks. PG is a diffusion planner-based method designed for long-horizon planning, where capturing temporally extended dependencies is essential. However, MuJoCo locomotion tasks involve short-horizon, dense-reward control focused on making agents run faster, which does not require such lookahead capabilities [1]. As a result, diffusion planner-based methods, including PG, often underperform compared to diffusion policy-based approaches in these settings, as shown in Table 1. While the additional structure and trajectory-level modeling of planners benefit challenging long-horizon domains such as AntMaze, Kitchen, or Maze2D, they provide little advantage and can even be slightly detrimental in simpler environments where policies can already achieve near-optimal performance.
>
> |                   | DQL        | DV*       | PG (planner) | PG (policy)        |
> |:-----------:|:--------------:|:-------------:|:----------------:|:----------------------:|
> | Walker2d-M  | **87.0**       | 79.5          | 82.3              | 84.5 $\pm$ 0.26         |
> | Walker2d-M-R| 95.5           | 83.5          | 83.7              | **96.4 $\pm$ 2.49**      |
> | Walker2d-M-E| **110.1**      | 109.0         | 109.4             | 109.8 $\pm$ 0.23        |
> | Hopper-M    | 90.5           | 84.1          | 97.5              | **98.4 $\pm$ 1.01**     |
> | Hopper-M-R  | **101.3**      | 91.3          | 91.3              | 100.5 $\pm$ 1.27        |
> | Hopper-M-E  | 111.1          | 109.9         | 110.4             | **111.4 $\pm$ 0.57**    |
> | HalfCheetah-M| 51.1          | 50.9          | 45.6              | **54.7 $\pm$ 0.42**     |
> | HalfCheetah-M-R| 47.8        | 46.4          | 46.4              | **48.6 $\pm$ 0.34**     |
> | HalfCheetah-M-E| **96.8**    | 92.3          | 95.2              | 95.5 $\pm$ 0.26         |
> | Average | 87.9       | 83.0          | 84.6              | **88.9**                |
>
> To examine whether the observed performance gap in MuJoCo tasks stems from the use of a diffusion planner rather than a limitation of the PG framework itself, we conduct an additional experiment by applying PG to a diffusion policy. Specifically, we replaced the standard Gaussian prior of a behavior-cloned diffusion policy with a learnable, behavior-regularized prior, following the same training procedure used in PG.
> The above table shows that replacing the diffusion planner in PG with a diffusion policy (PG policy) consistently improved performance across MuJoCo tasks, raising the average score from 84.6 (PG planner) to 88.9 and surpassing the DQL baseline score of 87.9. This demonstrates that PG remains effective even in short-horizon environments, suggesting that the lower performance of PG in MuJoCo tasks is due to the nature of trajectory-level planning rather than a limitation of the method itself. The results for PG (policy) are averaged over 3 random seeds and reported as the mean $\pm$ standard error.
>
> We will include these results to further clarify that the PG framework is effective across both policy- and planner-based approaches.
>
> ---
>
> **Q2. How does the method achieve good performance on long-horizon tasks, consistently outperforming MCSS? It seems that prior guidance alone does not provide this capability. Does the improvement come from Diffusion Veteran (DV)?**
>
> We apologize for the lack of clarity and appreciate the reviewer for raising this important point. We would like to clarify that in Figure 5, the method labeled as MCSS with 50 samples corresponds directly to Diffusion Veteran (DV). That is, DV is implemented by sampling 50 trajectories from a behavior-cloned diffusion planner and selecting the highest-value one using a learned value function. Importantly, all methods in Figure 5, including MCSS (with different sample counts) and PG, share the same diffusion model architecture and hyperparameters, all trained using the DV procedure. The performance differences arise solely from the inference-time guidance algorithms.
>
> While DV (i.e., MCSS-50) achieves strong performance, MCSS inherently couples the number of samples with the degree of behavior regularization. Increasing the number of samples improves the chance of finding higher-value trajectories, but it also raises the risk of selecting out-of-distribution ones, forcing a trade-off between trajectory quality and regularization strength. Because of this coupling, there is a practical ceiling on how precisely MCSS can optimize for the best high-value trajectory under a given level of regularization. Beyond a certain point, increasing samples no longer yields better results without sacrificing regularization.
>
> In contrast, PG completely decouples behavior regularization from the search for high-value trajectories. The $f$-divergence regularization is applied explicitly and independently, allowing the latent space prior to be optimized solely for maximizing trajectory value within the regularization constraint. This enables PG to accurately reach the behavior-regularized optimum, without approximation error from finite sampling, and without the computational cost or OOD risk that MCSS faces at large sample counts.
>
> We will revise the paper to clearly indicate that DV corresponds to MCSS with 50 samples, and that all methods share the same diffusion model. The observed improvements thus stem from our proposed prior-guided inference, rather than differences in model capacity or training setup.
>
> ---
>
> **Q3. It is unclear whether the method modifies the training of the base Diffusion Veteran. Is the proposed prior guidance applied only at the sampling stage?**
>
> We apologize for the confusion and we appreciate the reviewer’s careful reading. To clarify, PG does not modify the training of the underlying diffusion model used in Diffusion Veteran (DV). The diffusion planner is trained exactly as in DV, using standard behavior cloning on offline trajectories. PG is applied only during inference, where it replaces the MCSS procedure used in DV. Specifically:
>
> - During training we introduce an additional learnable prior distribution over the noise variable $\mathbf{x}_T$ that is optimized to maximize the expected value under the value function subject to an $f$‑divergence constraint maintaining closeness to the standard Gaussian.
> - This prior is trained after the diffusion model has been fixed through behavior cloning. At the sampling stage, the standard Gaussian prior is replaced with the learned prior distribution, after which the behavior‑cloned diffusion model is used for denoising.
> - The diffusion model itself is not fine-tuned or altered in any way.
>
> In summary, PG is a post‑hoc, inference‑time guidance method applied to a fixed diffusion model. All performance gains come from learned prior distribution rather than from altering the base diffusion model’s training. We will include this clarification in the next revision of the paper.
>
> ---
>
> **References**
>
> [1] Lu, Haofei, et al. "What makes a good diffusion planner for decision making?." *arXiv preprint arXiv:2503.00535* (2025).

---

> > ### Comment · Reviewer_J4nL · 2025-08-04
> >
> > I thank the authors for their detailed responses, which have addressed most of my concerns. I have no further questions at this time and will maintain my current evaluation. I will continue following the discussion of comments from the other reviewers.

---

### Official Review · Reviewer_tREs · 2025-07-02

**Clarity:** 4
**Significance:** 3
**Originality:** 3
**Rating:** 5
**Confidence:** 5

**Summary:**

This paper introduces Prior Guidance (PG) for diffusion planning for offline reinforcement learning. The idea is to steer a pretrained diffusion planner by learning a state-conditioned prior noise distribution that correspond to high-value trajectories. This avoids the need of multiple samples in Monte-Carlo search methods. Specifically, they parametrize the prior as a state-conditioned gaussian distribution, which is trained to maximize the (behavior) value function while staying close to the data distribution. By exploiting the bijective nature of the DDIM sampler and aliasing the sample-space value function to a noise-space value function, both objectives can be optimized in the prior space without backpropagating through the diffusion process. The paper evaluates PG on a suite of offline RL domains from the D4RL benchmark, where PG achieves strong performance compared to baselines, with a low computation cost.

**Questions:**

1. Why is the DDIM denoising process bijective in the limit of small discretization steps? Is there a prior work that discussed this?
2. Equation 2 does not explain what the variables are, e.g. $V$ and $f$. It would help to explain them immediately after the equation.
3. There is a recent related work on steering flow policies by learning the prior distribution [1]. They also train a noise-space value function to alias the sample-space value function. Can you discuss PG in relation to this work?

[1] Andrew Wagenmaker, Mitsuhiko Nakamoto, Yunchu Zhang, Seohong Park, Waleed Yagoub, Anusha Nagabandi, Abhishek Gupta, Sergey Levine. Steering Your Diffusion Policy with Latent Space Reinforcement Learning

**Ethical Concerns:**

["NO or VERY MINOR ethics concerns only"]

**Final Justification:**

The authors have sufficiently addressed my questions during the rebuttal. Overall, the paper proposes a practical method for finetuning diffusion models with RL, grounded on solid theoretical insights. Therefore I recommend acceptance.

**Limitations:**

The authors address the limitations of the paper, stating (1) additional training cost, (2) breaking the bijective mapping assumption at practical discretization steps, and (3) no image-based experiments as the main limitations.

**Quality:**

3

**Strengths And Weaknesses:**

**Strengths**
1. The paper presents a simple idea that can be applied to any pretrained diffusion planner -- just steer the diffusion planner by learning the prior distribution.
2. The mathematical derivations for the noise-space objectives (Equations 4 and 5) are sound.
3. The empirical results demonstrate state-of-the-art performance on the D4RL benchmark.
4. The ablation studies and analysis experiments are quite informative. For example, Figure 5 shows a clear tradeoff between optimality and value model exploitation as the number of samples in Monte-Carlo search grows. Figure 7 shows the correlation between the real value function and the aliased noise-space value function, revealing a mismatch in the bijective assumption (though empirically the policy performance is not affected).
5. The writing is clear and easy to follow.

**Weaknesses**
1. The D4RL results are not a significant improvement compared to the strongest baselines (DV*, DQL).

---

> ### Author Rebuttal · Authors · 2025-07-31
>
> **Dear Reviewer tREs,**
>
> We are grateful for your thoughtful feedback. We hope the following responses address your questions and help alleviate your concerns.
>
> ---
>
> **W1. The D4RL results are not a significant improvement compared to the strongest baselines (DV$^*$, DQL).**
>
> We appreciate the reviewer for their careful reading and insightful comment regarding PG’s performance. In Table 1 of the paper, PG shows substantial performance improvements over DQL and DV* in long-horizon tasks such as Kitchen, AntMaze, and Maze2D. However, in MuJoCo locomotion tasks such as Walker2d, Hopper, and HalfCheetah, the gains over DV* are modest, and the overall performance is lower than DQL. We attribute this not to a limitation of PG itself, but to the use of a diffusion planner, which is less effective in short-horizon, dense-reward settings where trajectory-level modeling provides little benefit and can even be slightly detrimental in simpler environments where policies can already achieve near-optimal performance.
>
> |             | **DQL**        | **DV***       | **PG (planner)** | **PG (policy)**        |
> |:-----------:|:--------------:|:-------------:|:----------------:|:----------------------:|
> | Walker2d-M  | **87.0**       | 79.5          | 82.3              | 84.5 $\pm$ 0.26         |
> | Walker2d-M-R| 95.5           | 83.5          | 83.7              | **96.4 $\pm$ 2.49**      |
> | Walker2d-M-E| **110.1**      | 109.0         | 109.4             | 109.8 $\pm$ 0.23        |
> | Hopper-M    | 90.5           | 84.1          | 97.5              | **98.4 $\pm$ 1.01**     |
> | Hopper-M-R  | **101.3**      | 91.3          | 91.3              | 100.5 $\pm$ 1.27        |
> | Hopper-M-E  | 111.1          | 109.9         | 110.4             | **111.4 $\pm$ 0.57**    |
> | HalfCheetah-M| 51.1          | 50.9          | 45.6              | **54.7 $\pm$ 0.42**     |
> | HalfCheetah-M-R| 47.8        | 46.4          | 46.4              | **48.6 $\pm$ 0.34**     |
> | HalfCheetah-M-E| **96.8**    | 92.3          | 95.2              | 95.5 $\pm$ 0.26         |
> | **Average** | **87.9**       | 83.0          | 84.6              | **88.9**                |
>
> To verify this, we conducted an additional experiment by applying PG to a diffusion policy instead of a planner. Specifically, we replaced the standard Gaussian prior of a behavior-cloned diffusion policy with a learnable, behavior-regularized prior, following the same training procedure used in PG. The above table shows that replacing the planner with a policy (PG policy) consistently improved performance across MuJoCo tasks, increasing the average score from 84.6 (PG planner) to 88.9, surpassing not only DV* (83.0) but also the DQL (87.9). This result demonstrates that PG remains effective even in short-horizon environments and the lower performance observed in MuJoCo tasks is due to the planner architecture rather than the PG framework itself. The results for PG (policy) are averaged over 3 random seeds and reported as the mean $\pm$ standard error.
>
> We will include these results to further clarify that the PG framework is effective across both policy- and planner-based approaches.
>
> ---
>
> **Q1. Why is the DDIM denoising process bijective in the limit of small discretization steps? Is there a prior work that discussed this?**
>
> We thank the reviewer for requesting this clarification. The DDIM denoising process [1] becomes bijective in the limit of small discretization steps because it corresponds to solving a particular ordinary differential equation (ODE) using Euler integration. In Section 4.3 of the DDIM paper, the authors show that DDIM sampling is equivalent to numerically solving an ODE of the form.
>
> $$
> d \bar{\mathbf{x}}(t)=\epsilon^{(t)}_\theta\bigg(\frac{\bar{\mathbf{x}}(t)}{\sqrt{\sigma(t)^2 +1}}\bigg) \cdot d \sigma(t),
> $$
>
> where $\bar{\mathbf{x}}(t)=\frac{\mathbf{x}(t)}{\sqrt{\bar{\alpha}_t}}$ and $\sigma(t) = \sqrt{\frac{1-\bar{\alpha}_t}{\bar{\alpha}_t}}$. This ODE describes a deterministic transformation from noise $\bar{\mathbf{x}}(T)$  to data $\bar{\mathbf{x}}(0)$, and vice versa. Therefore, with sufficiently fine discretization, the DDIM process approximates the ODE solution arbitrarily well. Furthermore, Section 4.3 and Appendix B of [1] explicitly note that DDIM corresponds to the probability flow ODE for the variance-exploding SDE from Song et al. (2020) [2].
>
> This equivalence explains why the DDIM sampling path becomes bijective when using small discretization steps. Several works leveraged this property for DDIM inversion, the process of mapping real data samples back into the latent noise space [1, 3, 4].
>
> ---
>
> **Q2. Equation 2 does not explain what the variables are, e.g $V$ and $f$. It would help to explain them immediately after the equation.**
>
> We thank the reviewer for pointing out the lack of variable definitions in Equation (2). We agree that the meanings of $V$ and $f$ should be clarified immediately following the equation to improve readability, as the original explanation in the Preliminary section is far from Equation (2) and may disrupt the reader’s flow. To address this, we will revise the main paper to include explicit definitions right after Equation (2). Specifically, we will add the following sentence in the next revision of the paper:
>
> Here, $V(\mathbf{x}_0)$ denotes the estimated value of the denoised trajectory $\mathbf{x}_0$ and $f(\cdot)$ is a regularization function (e.g., for $f$-divergence) that penalizes deviation from the behavior distribution.
>
> ---
>
> **Q3. There is a recent related work on steering flow policies by learning the prior distribution [1]. They also train a noise-space value function to alias the sample-space value function. Can you discuss PG in relation to this work?**
>
> We appreciate the reviewer for bringing up this connection. A referenced concurrent work, "Steering Your Diffusion Policy with Latent Space Reinforcement Learning" (DSRL) [5], proposes steering a behavior-cloned diffusion policy by optimizing a learnable prior via reinforcement learning. While the high-level goal of guiding generative models through learned priors is shared with our method, there are both important commonalities and key differences:
>
> **Commonalities: Training Latent Prior and Value function**
>
> Both DSRL and PG aim to improve decision quality by learning the prior over the noise space ($\mathbf{x}_T \sim \mathcal{N}(\mathbf{0}, \mathbf{I})$) that feeds into the diffusion model’s denoising process, allowing for the sampling of high-value actions or trajectories. In both methods, the prior is trained to maximize the value function defined over $\mathbf{x}_T$, which estimates the value of the corresponding denoised sample $\mathbf{x}_0$.
>
> **Key Differences**
>
> 1. **Diffusion Planner and Diffusion Policy**
>
>     Unlike DSRL, which  employs a diffusion policy that directly maps noise to action, PG utilizes a diffusion planner that outputs entire state trajectories and uses an inverse dynamics model to reconstruct executable actions. This makes PG particularly suitable for long-horizon and sparse-reward settings, where trajectory-level modeling is essential.
>
> 2. **Theoretical Insight**
>
>     In our work, we provide a formal result (Proposition 1) demonstrating that, with a sufficiently large number of discretization steps, the objective for optimizing the diffusion model can be equivalently reformulated as an optimization over the latent prior distribution. This equivalence is enabled by the bijective nature of the DDIM sampling process, which allows for an exact mapping between the data space and the latent prior. This reformulation is not explored in DSRL.
>
> 3. **Handling OOD Samples in Offline RL**
>
>     DSRL assumes that the learned diffusion model generates only in‑distribution samples and therefore does not incorporate explicit behavior constraints. However, as shown in Figures 3 and 5, diffusion models trained on offline datasets can still produce out‑of‑distribution samples that degrade performance. To mitigate this issue, PG introduces behavior regularization by imposing an $f$‑divergence penalty between the learned prior and the standard normal distribution. This explicitly penalizes deviation from the behavior distribution and induces regularization at the prior level.
>
>
> We will revise the Related Works section to clearly position our contributions relative to DSRL and highlight both the shared foundations and the novel aspects of our approach.
>
> ---
>
> **References**
>
> [1] Song, Jiaming, Chenlin Meng, and Stefano Ermon. "Denoising diffusion implicit models." *arXiv preprint arXiv:2010.02502* (2020).
>
> [2] Song, Yang, et al. "Score-based generative modeling through stochastic differential equations." *arXiv preprint arXiv:2011.13456* (2020).
>
> [3] Dhariwal, Prafulla, and Alexander Nichol. "Diffusion models beat gans on image synthesis." Advances in neural information processing systems 34 (2021): 8780-8794.
>
> [4] Zeng, Yan, Masanori Suganuma, and Takayuki Okatani. "Inverting the generation process of denoising diffusion implicit models: Empirical evaluation and a novel method." *2025 IEEE/CVF Winter Conference on Applications of Computer Vision (WACV)*. IEEE, 2025.
>
> [5] Wagenmaker, Andrew, et al. "Steering Your Diffusion Policy with Latent Space Reinforcement Learning." arXiv preprint arXiv:2506.15799 (2025).

---

> > ### Comment · Reviewer_tREs · 2025-08-04
> >
> > Thank you for addressing my comments and questions. The discussion in relation to DSRL offers plenty of insights, and including it in the revised manuscript will better position this paper in the literature. I have no further question and will maintain my evaluation of the paper.

---

### Decision · Program_Chairs · 2025-09-17

**Decision:**

Accept (poster)

**Comment:**

This paper proposes Prior Guidance (PG) for diffusion-based planning in offline RL. Instead of using a fixed Gaussian prior, PG introduces a learnable, state-conditioned prior optimized with a behavior-regularized objective. The method avoids costly Monte Carlo sampling at inference time by training a latent-space value function that aliases the denoised trajectory value, thereby steering the diffusion process efficiently. The approach is evaluated on D4RL tasks, showing strong results in long-horizon environments (AntMaze, Kitchen, Maze2D) and competitive performance on others.

Reviewers agreed that the paper is clearly written, the idea is simple yet elegant, and the derivations are sound. Strengths include: (i) computational efficiency by replacing sampling-heavy MCSS, (ii) novel reformulation of guidance in latent space, (iii) tractable regularization, and (iv) comprehensive ablations and analysis.
Concerns were mainly empirical. Multiple reviewers noted that PG underperforms on short-horizon MuJoCo locomotion tasks compared to strong diffusion policies such as DQL. This raised questions about whether the benefits of PG are limited to long-horizon planning. The rebuttal addressed this point with additional experiments: applying PG to a diffusion policy (rather than a planner) improved performance on MuJoCo tasks, surpassing DQL. This clarified that the performance gap stems from the planner architecture, not PG itself. Reviewers also raised questions on assumptions (e.g., bijectivity of DDIM), the use of a simple Gaussian prior, and the reliance on DV as a base model. The authors clarified that bijectivity is approximate but sufficient in practice, PG can easily be extended to more expressive priors, and ablations confirm PG provides gains independently of DV. Clarifications on Kitchen tasks, the inverse dynamics model, and connections to concurrent work (DSRL) were also promised.

Overall, this is a technically solid and well-presented contribution. While performance gains are modest in some saturated benchmarks, PG clearly improves efficiency and shows notable advantages in long-horizon settings. The rebuttal convincingly addressed most concerns. Based on the reviewers’ feedback, I recommend that the authors include the additional ablation studies conducted during the rebuttal on applying PG to a diffusion policy, and clearly articulate the appropriate application scope of PG. In addition, the authors should provide a clearer summary of the experimental results to better highlight the benefits of PG.